# Glucose-Binding of Periplasmic Protein GltB Activates GtrS-GltR Two-Component System in *Pseudomonas aeruginosa*

**DOI:** 10.3390/microorganisms9020447

**Published:** 2021-02-21

**Authors:** Chenchen Xu, Qiao Cao, Lefu Lan

**Affiliations:** 1University of Chinese Academy of Sciences, No.19A Yuquan Road, Beijing 100049, China; 201528012342123@simm.ac.cn; 2State Key Laboratory of Drug Research, Shanghai Institute of Materia Medica, Chinese Academy of Sciences, Shanghai 201203, China; caoqiao@simm.ac.cn; 3School of Pharmaceutical Science and Technology, Hangzhou Institute for Advanced Study, UCAS, Hangzhou 310024, China; 4NMPA Key Laboratory for Testing Technology of Pharmaceutical Microbiology, Shanghai Institute for Food and Drug Control, Shanghai 201203, China

**Keywords:** *Pseudomonas aeruginosa*, Glucose, periplasmic binding protein, two-component system, transcriptional regulation

## Abstract

A two-component system GtrS-GltR is required for glucose transport activity in *P. aeruginosa* and plays a key role during *P. aeruginosa*-host interactions. However, the mechanism of action of GtrS-GltR has not been definitively established. Here, we show that *gltB*, which encodes a periplasmic glucose binding protein, is essential for the glucose-induced activation of GtrS-GltR in *P. aeruginosa*. We determined that GltB is capable of binding to membrane regulatory proteins including GtrS, the sensor kinase of the GtrS-GltR TCS. We observed that alanine substitution of glucose-binding residues abolishes the ability of GltB to promote the activation of GtrS-GltR. Importantly, like the *gtrS* deletion mutant, *gltB* deletion mutant showed attenuated virulence in both *Drosophila melanogaster* and mouse models of infection. In addition, using CHIP-seq experiments, we showed that the promoter of *gltB* is the major in vivo target of GltR. Collectively, these data suggest that periplasmic binding protein GltB and GtrS-GltR TCS form a complex regulatory circuit that regulates the virulence of *P. aeruginosa* in response to glucose.

## 1. Introduction

Glucose is an essential nutrient for the human body, serving as a primary fuel for energy production [1]. Many studies have also suggested that pathogens can exploit glucose as either a nutrient or a signal molecule that induces virulence functions [2,3,4,5,6,7,8,9,10]. Indeed, glucose acquisition is essential for the full virulence potential of pathogenic bacteria [11,12] including *Pseudomonas aeruginosa*, a major cause of nosocomial infection as well as infection in hosts with compromised defenses [13,14]. Recently, the World Health Organization has categorized *P. aeruginosa* as a critical priority pathogen that is in urgent need of new antibiotics [15,16]. Establishing a deeper understanding of how *P. aeruginosa* senses and responds to glucose may allow for the rational design of improved therapies to treat or prevent infections caused by this formidable pathogen [17,18,19].

It has been proposed that, in *P. aeruginosa*, when glucose passes the outer membrane through the OprB porin and reaches the periplasm, it can be transported by an ABC transport system (i.e., ABC^GltBFGK^, also designated as ABC^GtsABCD^) to the cytoplasm, or it can be oxidized in the periplasm by glucose dehydrogenase (Gcd) and gluconate dehydrogenase (Gad) to gluconate and 2-ketogluconate (2-KG) [20,21,22,23,24] (Figure 1A). Gluconate and 2-KG are subsequently transported to the cytoplasm via transporters GntP and KguT, respectively [22,25] (Figure 1A). Both the phosphorylative pathway, which converts glucose to 6-phosphogluoconate by the combined action of glucokinase (GlK), glucose 6-phosphate dehydrogenase (Zwf), and 6-phosphogluconolactonase (Pgl) (Figure 1A), and the direct oxidative pathway, wherein glucose is oxidized to gluconate, 2-ketogluconate and then subsequently to 6-phosphogluconate (Figure 1A), are active under aerobic conditions, while the phosphorylative pathway can substitute for the oxidative pathway under anaerobic conditions [21].

Genes encoding the *P. aeruginosa* glucose catabolism are organized in operons and are under the control of different regulators (i.e., HexR, PtxS, PtxR, GntR, and GtrS-GltR) [23]. Among them, response regulator (RR) GltR, which forms a two-component system with its cognate sensor kinase (SK) GtrS [26], is required for glucose transport activity in *P. aeruginosa* PAO1 [20,27], presumably by activating the transcription of *gltBFGK* (*gtsABCD*)-*oprB* operon that encodes OprB porin and the glucose transport system ABC^GltBFGK^ [28]. However, these results were somehow contrast to the findings of Daddaoua et al. [26], where GltR is a transcriptional repressor that is released from its target operators of glucose metabolism genes upon phosphorylation.

Although discrepancy exists, results from previous studies show a conserved role of GtrS-GltR on glucose utilization in different *Pseudomonas* species including *P. aeruginosa* [20,26,29]. Interestingly, 2-KG and 6-phosphogluconate (6PG), two metabolic intermediates of glucose utilization pathways, can bind to ligand-binding domain of GtrS, leading to increase GtrS autophosphorylation and GltR phosphorylation activity [26]. These observations suggest a physiological relevance of GtrS-GltR in glucose metabolism. Importantly, GtrS plays a key role during *P. aeruginosa*-host interactions and is required for optimal colonization and dissemination in a mouse model of infection [28]. Intrigued by these findings, in this study, we decided to investigate the mechanism of action of GtrS-GltR TCS in *P. aeruginosa* PAO1.

## 2. Experimental Procedures

### 2.1. Bacterial Strains, Plasmids, and Culture Conditions

The bacterial strains and plasmids used in this study are listed in Appendix A. Unless noted otherwise, *E. coli* cultures were grown in Luria-Bertani (LB) medium. *P. aeruginosa* PAO1 [19] and its derivatives were grown in LB medium, Pyocyanin production broth (PB medium) (20 g peptone, 1.4 g MgCl_2_, 10 g K_2_S_4_, 20 mL glycerol per liter; pH 7.0) [30], minimal medium (MM) (6.8 g Na_2_HPO_4_, 3 g KH_2_PO_4_, 0.5 g NaCl, 0.24 g MgSO_4_, 1 g NH_4_Cl; pH 7.4) supplemented with 10 mM glucose, gluconate or 2-KG as the sole carbon source, M8 minimal medium (MM) (6.8 g Na_2_HPO_4_, 3 g KH_2_PO_4_, 0.5 g NaCl, 0.24 g MgSO_4_, 0.82 g sodium acetate, 0.5 g glutamate per liter; pH 7.4), or M9 minimal medium (MM) (6.8 g Na_2_HPO_4_, 3 g KH_2_PO_4_, 0.5 g NaCl, 0.24 g MgSO_4_, 1 g NH_4_Cl, 0.96 g citrate, 2.5 mL of A9 solution; pH 7.4) [25,31]. A9 solution was composed of the following (in milligrams per liter): HBO_3_, 300; ZnCl_2_, 50; MnCl_2_∙4H_2_O, 30; CoCl_2_, 200; CuCl_2_∙2H_2_O, 10; NiCl_2_∙6H_2_O, 20; and NaMoO_4_∙2H_2_O, 30. Bacterial cultures were incubated at 37 °C with shaking (250 rpm) under aerobic condition. For plasmid maintenance, antibiotics were used at the following concentrations where appropriate: for *E. coli*, carbenicillin at 100 μg/mL, kanamycin at 50 μg/mL, tetracycline at 5 μg/mL, and gentamicin at 10 μg/mL; for *P. aeruginosa*, gentamicin at 50 μg/mL in LB or 150 μg/mL in *Pseudomonas* Isolation Agar (PIA; BD); tetracycline at 30 μg/mL in LB or 150 μg/mL in PIA; carbenicillin at 150 μg/mL in LB, PIA, and minimal medium.

### 2.2. Construction of P. aeruginosa Mutants

For gene replacement, a SacB-based strategy was employed as previously described [32]. To generate the Δ*gltB* mutant strain, polymerase chain reactions (PCRs) were performed in order to amplify sequences upstream (~1.2 kb) and downstream (~1.3 kb) of the intended deletion. The upstream fragment was amplified from PAO1 genomic DNA using D-*gltB*-up-F (with *Eco*RI site) and D-*gltB*-up-R (with *Kpn*I site), while the downstream fragment was amplified from PAO1 genomic DNA using D-*gltB*-down-F (with *Kpn*I site) and D-*gltB*-down-R (with *Hin*dIII site). The two products were digested and then cloned into the *Eco*RI/*Hin*dIII-digested gene replacement vector pEX18Ap, yielding pEX18Ap::*gltB*UD. A 1.8 kb gentamicin resistance cassette was cut from pPS858 with *Kpn*I and then cloned into pEX18Ap::*gltB*UD, yielding pEX18Ap::*gltB*UGD. The resultant plasmid was electroporated into PAO1 with selection for gentamicin resistance. Colonies were screened for gentamicin resistance and loss of sucrose (5%) sensitivity, which indicates a double-cross-over event and the gene replacement. The deletion of *gltB* was confirmed by PCR.

A similar method was used to generate the Δ*gtrS*, Δ*gtrS-gltR*, Δ*gcd*, Δ*gad* strains. Briefly, for the construction of Δ*gtrS*, the upstream fragment (~1 kb) of the intended deletion was amplified with primers D-*gtrS*-up-F (with *Eco*RI site) and D-*gtrS*-up-R (with *Bam*HI site), and the downstream fragment (~1 kb) was amplified with primer pair D-*gtrS*-down-F (with *Bam*HI site)/D-*gtrS*-down-R (with *Hin*dIII site). For the construction of Δ*gtrS-gltR*, the upstream fragment (~1.3 kb) of the intended deletion was amplified with primer pair D-*gltR*-up-F/D-*gltR*-up-R (*Eco*RI/*Bam*HI sites), while the downstream fragment (~1 kb) was amplified with primer pair D-*gtrS*-down-F/D-*gtrS*-down-R (*Bam*HI/*Hin*dIII). For the construction of Δ*gcd*, the upstream fragment (~1.1 kb) of the intended deletion was amplified with primers D-*gcd*-up-F (with *Eco*RI site) and D-*gcd*-up-R (with *Bam*HI site), and the downstream fragment (~1.1 kb) was amplified with primers D-*gcd*-down-F (with *Bam*HI site) and D-*gcd*-down-R (with *Hin*dIII site). For the construction of Δ*gad*, the upstream fragment (~1 kb) of the intended deletion was amplified with primers D-*gad*-up-F (with *Eco*RI site) and D-*gad*-up-R (with *Bam*HI site), and the downstream fragment (~1.1 kb) was amplified with primers D-*gad*-down-F (with *Bam*HI site) and D-*gad*-down-R (with *Hin*dIII site). A 1.8 kb gentamicin resistance cassette was cut from pPS858 and then cloned into the pEX18Ap::*gtrS*UD, pEX18Ap::*gtrS-gltR*UD, pEX18Ap::*gcd*UD and pEX18Ap::*gad*UD, yielding pEX18Ap::*gtrS*UGD, pEX18Ap::*gtrS-gltR*UGD, pEX18Ap::*gcd*UGD, and pEX18Ap::*gad*UGD, respectively, as described above. To generate Δ*gcd*Δ*gtrS-gltR* mutant, the gentamicin resistance cassette of Δ*gtrS-gltR* was excised by using the plasmid pFLP2 that encoded Flp recombinase [32], and then the pEX18Ap::*gcd*UGD plasmid (Appendix A) was electroporated into the Δ*gtrS-gltR* (without gentamicin resistance cassette) in order to generate Δ*gcd*Δ*gtrS-gltR*. All the primers used for PCRs are listed in Appendix A. All the mutant strains were confirmed by PCR.

### 2.3. Construction of Plasmids

All the primers used for plasmid construction are listed in Appendix A. For constructing pAK1900-*gltB* (p-*gltB*), a ~1.3 kb PCR product covering 40 bp of the *gltB* upstream region, the *gltB* gene, and 15 bp downstream of *gltB* was amplified from PAO1 genomic DNA using primers *gltB*-comp-F (with *Hin*dIII site) and *gltB*-comp-R (with *Kpn*I site). For generating pAK1900-*gtrS* (p-*gtrS*), a ~1.5 kb PCR product covering 72 bp of the *gtrS* upstream region, the *gtrS* gene, and 9 bp downstream of *gtrS* was amplified from PAO1 genomic DNA using primers *gtrS*-comp-F (with *Hin*dIII site) and *gtrS*-comp-R (with *Bam*HI site). For generating pAK1900-*gtrS-gltR* (p-*gtrS-gltR*), a ~2.3 kb PCR product covering 38 bp of the *gltR* upstream region, the *gltR* and *gtrS* genes, and 9 bp downstream of *gtrS* was amplified from PAO1 genomic DNA using primers *gltR*-comp-F (with *Hin*dIII site) and *gtrS*-comp-R (with *Bam*HI site). To generate pAK1900-*gcd* (p-*gcd*), a ~2.4 kb PCR product covering 30 bp of the *gcd* upstream region, the *gcd* gene, and 8 bp downstream of *gcd* was amplified from PAO1 genomic DNA using primers *gcd*-comp-F (with *Hin*dIII site) and *gcd*-comp-R (with *Bam*HI site). All the products were digested with the indicated enzymes and cloned into pAK1900 [33], and the direction of transcription of the cloned genes is in the same orientation as p*lac* on pAK1900.

To construct pAK1900-*gltB*^W33A^ (p-*gltB*^W33A^; *gltB*^W33A^, the tryptophan 33 of GltB was mutated to alanine), the primer pair GltB (W33A)-F/GltB (W33A)-R and a QuikChange II site-directed mutagenesis kit (Stratagene, Catalog#:200518) were used. A similar method was used to generate the pAK1900-*gltB*^W34A^ (p-*gltB*^W34A^; *gltB*^W34A^, the tryptophan 34 of GltB was mutated to alanine), pAK1900-*gltB*^K90A^ (p-*gltB*^K90A^; *gltB*^K90A^, the lysine 90 of GltB was mutated to alanine), pAK1900-*gltB*^W268A^ (p-*gltB*^W268A^; *gltB*^W268A^, the tryptophan 268 of GltB was mutated to alanine), pAK1900-*gltB*^D301A^ (p-*gltB*^D301A^; *gltB*^D301A^, the asparticacid 301 of GltB was mutated to alanine). The primer pair GltB (W34A)-F/GltB (W34A)-R was used to generate p-*gltB*^W34A^. The primer pair GltB (K90A)-F/GltB (K90A)-R was used to construct p-*gltB*^K90A^. The primer pair GltB (W268A)-F/GltB (W268A)-R was used to generate p-*gltB*^W268A^. The primer pair GltB (D301A)-F/GltB (D301A)-R was used to construct p-*gltB*^D301A^.

For generating pAK1900-*gtrS*-YFP (p-*gtrS-*YFP), primers *gtrS*-comp-F (with *Hin*dIII site) and *gtrS*-Y-R (with *Bam*HI site) were used to perform PCR of the *gtrS* gene that was meant to fuse with a C-terminal YFP-tag. A ~1.5 kb PCR product covering 72 bp upstream of *gtrS* (not including the stop codon) was generated from PAO1 genomic DNA using primers *gtrS*-comp-F (with *Hin*dIII site) and *gtrS*-Y-R (with *Bam*HI site) and cloned into the pAK1900. A ~0.75 kb PCR product covering yellow fluorescent protein (YFP) fragment was amplified from plasmid pEYFP (Appendix A) using primers YFP-F (with *Bam*HI site) and YFP-R (with *Kpn*I site). The YFP PCR product was digested and then cloned into *Bam*HI/*Kpn*I-digested plasmid p-*gtrS*, yielding p-*gtrS*-YFP. To construct pAK1900-*gtrS*^H280A^ (p-*gtrS*^H280A^; *gtrS*^H280A^, the histidine 280 of GtrS was mutated to alanine) and pAK1900-*gtrS*^H280A^-YFP (p-*gtrS*^H280A^-YFP), the primer pair GtrS (H280A)-F/GtrS (H280A)-R and a QuikChange II site-directed mutagenesis kit (Stratagene, Catalog#:200518) were used.

To construct pAK1900-*gltR*-flag (p-*gltR*-flag), a ~0.7 kb PCR product covering 38 bp upstream of *gltR* (not including the stop codon) was amplified from PAO1 genomic DNA using primers *gltR*-comp-F (with *Hin*dIII site) and *gltR*-flag-R (with *Bam*HI site).

All constructs were sequenced to ensure that no unwanted mutations resulted.

### 2.4. Transposon Mutagenesis

The *gltB* promoter region (−740 to +20 of the start codon) was amplified by PCR using the primers pro-*gltB*-F (with *Xho*I site) and pro-*gltB*-R (with *Bam*HI site) and cloned into the *Xho*I and *Bam*HI sites of the mini-CTX-*lacZ* [34] to generate mini-CTX-*gltB-lacZ*. The construct was sequenced to ensure that no unwanted mutations resulted. The resulting plasmid was conjugated into wild type PAO1 strain and the construct was integrated into the *attB* site as described previously [32], through a diparental mating using *E. coli* S17 λ-pir as the donor. The tetracycline resistance cassette wasdeleted using a flippase (FLP) recombinase encoded on the pFLP2 plasmid [32].

For transposon mutagenesis, the transposon in pBT20 [35] was conjugally transferred by biparental mating into PAO1/mini-*gltB-lacZ*, following a protocol previously described [36]. Briefly, the donor strain (*E. coli* SM10-λ *pir*) containing the pBT20 and the recipient PAO1/mini-*gltB-lacZ* strain were scraped from overnight plates and suspended in 1 mL of buffer (6.8 g Na_2_HPO_4_, 3 g KH_2_PO_4_, 0.5 g NaCl). Concentrations of the bacteria in the suspensions were adjusted to OD_600_ of 40 for the donor and OD_600_ of 20 for the recipient. Next, each donor and recipient were mixed together and spotted on a LB agar plate and incubated at 37 °C for 6 h. Mating mixtures were scraped and resuspended in 1 mL of buffer. Transposon-mutagenized bacteria were selected by plating on PIA plates containing gentamicin at 150 µg/mL. A sterile tip was used to pick up individual colonies and dip them into the M8 minimal agar plates (supplying 1.5% agar and 10 mM glucose) with 20 µg/mL 5-bromo-4-chloro-3-indolyl-β-D-galactoside (X-gal). Approximately 50,000 colonies were screened for the appearance of blue color. The localization of the Mariner transposon with respect to the *P. aeruginosa* genome was determined using an established protocol [35].

### 2.5. Expression of the Recombinant Proteins in E. coli and their Purification

The following seven recombinant proteins were expressed in *E. coli*: (1) 6His-GltB, the N-terminal 6His-tagged GltB; (2) 6His-GltB^D301A^, a 6His-GltB variant in which the aspartic acid 301 of GltB was replaced by alanine; (3) 6His-GtrS^LBD^, the N-terminal 6His-tagged ligand-binding domain of GtrS (residues 29-199); (4) 6His-GltR, the N-terminal 6His-tagged GltR; (5) 6His-GltR^D56A^, a 6His-GltR variant in which the aspartic acid 56 of GltR was replaced by alanine; (6) 6His-PctA^LBD^, the N-terminal 6His-tagged ligand-binding domain of PctA (residues 30-278); (7) 6His-PA2788^LBD^, the N-terminal 6His-tagged ligand-binding domain of PA2788 (residues 44-179).

For construction of the expression plasmids, the corresponding DNA fragments were amplified by PCR using the primers listed in Appendix A. Briefly, primer pairs *gltB*-F/*gltB*-R (*Nde*I/*Xho*I), *gtrS*-LBD-F/*gtrS*-LBD-R (*Nde*I/*Bam*HI), *gltR*-F/*gltR*-R (*Bam*HI/*Xho*I), *pctA*-LBD-F/*pctA*-LBD-R (*Bam*HI/*Hin*dIII), and *pa2788*-LBD-F/*pa2788*-LBD-R (*Bam*HI/*Hin*dIII) were, respectively, used to perform PCR of *P. aeruginosa* PAO1 *gltB, gtrS*^LBD^, *gltR*, *pctA*^LBD^ and *pa2788*^LBD^ genes. The PCR products were digested with corresponding restriction endonuclease and cloned into pET28a, yielding pET28a-*gltB*, pET28a-*gtrS*^LBD^, pET28a-*gltR*, pET28a-*pctA*^LBD^, and pET28a-*pa2788*^LBD^, respectively. The resulting plasmids allowed the expression of the fusion protein with a 6His-tag at their N-terminus. Colonies were verified by sequencing.

To generate 6His-GltB^D301A^ and 6His-GltR^D56A^, the pairs of partially overlapping mutagenic primers GltB^D301A^-F/GltB^D301A^-R and GltR^D56A^-F/GltR^D56A^-R were, respectively, used to amplify the entire plasmid with a high-fidelity DNA polymerase. Then, the template DNA was eliminated by digestion with *Dpn*I. The resulting products were transformed into *E. coli* DH5α and colonies were selected on LB supplemented with kanamycin (50 μg/mL). Plasmid from resulting clones was isolated and verified by sequencing.

The protein was expressed in *E. coli* strain BL21 star (DE3) and purifications were performed as previously described [36,37]. Briefly, bacteria were grown at 37 °C overnight in 10 mL of LB medium with shaking (250 rpm). The cultures were transferred into 1 L of LB medium supplemented with kanamycin (50 μg/mL) incubated at 37 °C with shaking (200 rpm) until the OD_600_ reached 0.6, and then isopropyl-1-thio-β-d-galactopyranoside (IPTG) was added to a final concentration of 1 mM. Then, the cultures were incubated overnight at 16 °C with shaking (200 rpm). The cultures were harvested by centrifugation at 8000 rpm for 8 min and then cells were stored at −80 °C.

The cells were suspended in 40 mL of buffer A [50 mM Tris-HCl, PH 7.5, 300 mM NaCl, 10 mM imidazole, 1 mM dithiothreitol (DTT)] and lysed at 4 °C by sonication to produce lysates containing all of the cell contents. The whole cell fraction was subjected to centrifugation at 4 °C at 12,000 rpm for 25 min to remove insoluble material and the membrane fraction. The resulting supernatant was loaded onto a 5 mL His-Trap column (Code#: 17-5247-01, GE Healthcare), equilibrated with buffer A and eluted with a 0–100% gradient of buffer B (50 mM Tris-HCl, pH 7.5, 300 mM NaCl, 500 mM imidazole, 1 mM DTT). After that, the fractions containing 6His-GltB, 6His-GtrS^LBD^, 6His-GltR, 6His-PctA^LBD^, 6His-PA2788^LBD^, 6His-GltB^D301A^ or 6His-GltR^D56A^ were collected and loaded onto the HiTrap Desalting 5 × 5 mL (Sephadex G-25 S) (Code#: 17-1408-01, GE Healthcare) and eluted with buffer R (50 mM Tris-HCl, PH 7.5, 300 mM NaCl, 1 mM DTT) to remove the imidazole. The purified protein was >90% pure as estimated by a 12% (wt/vol) SDS/PAGE gel. Protein concentrations were determined using the Nanodrop 2000 by A280.

### 2.6. Monitoring Gene Expression by Lux-Based Reporters

The plasmid mini-CTX-lux (carrying a promoterless *luxCDABE* reporter gene cluster, Appendix A) [34] was used to construct promoter-*luxCDABE* reporter fusions *gltB*-*lux*. The *gltB* promoter region (−740 to +20 of the start codon) was amplified by PCR using the primers pro-*gltB*-F (with *Xho*I site) and pro-*gltB*-R (with *Bam*HI site) and cloned into the *Xho*I and *Bam*HI sites of the mini-CTX-lux (Appendix A) to generate mini-CTX-*gltB-lux*. The resulting plasmid was conjugated into *P. aeruginosa* strains and the construct was integrated into the *attB* site as described previously [32] through a diparental mating using *E. coli* S17 λ-pir as the donor.

A similar method was used to generate *gntR-lux*, *ptxS-lux, glk-lux*, *edd-lux*, *toxA-lux*, *opgG-lux*, and *ctpH-lux*. The *gntR* promoter region (−466 to +31 of the start codon) was amplified by PCR using the primers pro-*gntR*-F (with *Hin*dIII site) and pro-*gntR*-R (with *Bam*HI site). The *ptxS* promoter region (−749 to +27 of the start codon) was amplified by PCR using the primers pro-*ptxS*-F (with *Hin*dIII site) and pro-*ptxS*-R (with *Eco*RI site). The *glk* promoter region (−730 to +56 of the start codon) was amplified by PCR using the primers pro-*glk*-F (with *Hin*dIII site) and pro-*glk*-R (with *Bam*HI site). The *edd* promoter region (−520 to +26 of the start codon) was amplified by PCR using the primers pro-*edd*-F (with *Hin*dIII site) and pro-*edd*-R (with *Bam*HI site). The *toxA* promoter region (−980 to −5 of the start codon) was amplified by PCR using the primers pro-*toxA*-F (with *Hin*dIII site) and pro-*toxA*-R (with *Bam*HI site). The *opgG* promoter region (−493 to +20 of the start codon) was amplified by PCR using the primers pro-*opgG*-F (with *Hin*dIII site) and pro-*opgG*-R (with *Bam*HI site). The *ctpH* promoter region (−229 to +99 of the start codon) was amplified by PCR using the primers pro-*ctpH*-F (with *Hin*dIII site) and pro-*ctpH*-R (with *Bam*HI site). To construct the plasmid *gltB-D-lux* (*gltB-lux* which lacking the conserved GltR-binding site, 5′-GTGACAAA-3′), the primer pair pro-*gltB-D*-F/pro-*gltB-D*-R and a QuikChange II site-directed mutagenesis kit (Stratagene, Catalog#:200518) were used. In PAO1 and its derivatives, parts of the mini-CTX-lux vector containing the tetracycline resistance cassette were deleted using a flippase (FLP) recombinase encoded on the pFLP2 plasmid. All the promoters are oriented in the same direction as *luxCDABE*, and constructs were sequenced to ensure that no unwanted mutations resulted.

Unless noted otherwise, the expression of promoter fusion genes was measured in a 96-well black-wall clear-bottom plate (Corning incorporated, Costar, Code#:3603). Briefly, *P. aeruginosa* strains were grown in M8 minimal medium at 37 °C overnight with shaking at 250 rpm, then diluted 50-fold in 1 mL of fresh M8 minimal medium supplemented with or without 10 mM glucose. A 150 μL volume of the sample was added to the wells and a 80 μL volume of filter-sterilized mineral oil was further added in order to prevent evaporation during the assay. Promoter activities were measured as counts per second (CPS) of light production with a Synergy 2 Multi-Mode Microplate Reader as described previously [36,38]. Relative light units were calculated by normalizing CPS to OD_600_.

The expression of promoter fusion genes was also carried out using a tube culture method, as indicated. Strains were grown in M9 minimal medium [31] in the presence or absence of 5 mM effector (glucose, gluconate or 2-ketogluconate), as indicated. Overnight cultures were diluted to a turbidity of 0.01 in the fresh M9 minimal medium supplemented with or without effector. The diluted cultures were grown in a 20 mL tube with a tube volume-to-medium volume ratio of 5:1 and growth was continued at 37 °C with shaking (250 rpm) for 8 h, a 100 μL volume of the sample was added to the well of a 96-well black-wall clear-bottom plate (Corning incorporated, Costar, Code#: 3603) in order to measure the CPS of light production. Each sample was tested in triplicate. Relative light units were calculated by normalizing CPS to OD_600_.

### 2.7. Bacterial Growth Assays

*P. aeruginosa* strains were grown in LB supplemented with carbenicillin (100 μg/mL) at 37 °C overnight with shaking at 250 rpm. Overnight cultures were washed thrice. Then, the cultures were diluted 100-fold in 1 mL of fresh minimal medium (MM) supplemented with 10 mM glucose as the sole carbon source. After that, a 150 μL volume of the sample was added to the wells and an 80 μL volume of filter-sterilized mineral oil was added in order to prevent evaporation during the assay. The absorption of OD_600_ was detected at different time points at 37 °C by a Synergy 2 Multi-Mode Microplate Reader as described previously [36,38]. A similar experimental procedure was used to detect bacterial growth on 10 mM gluconate or 2-ketogluconate (2-KG) as the sole carbon source.

### 2.8. Thermal Shift Assays

Thermal Shift assays were performed as previously described [39] with some modifications. Reaction mixtures were prepared by 5 μM purified proteins and molecule to a desired concentration into the buffer A (50 mM Tris-HCl, PH 7.5, 300 mM NaCl, 10 mM imidazole, 1 mM DTT) and 2 μL of 50 × Sypro Orange (Invitrogen, Lot: 1873204) was added to the mixture to yield a final reaction volume of 20 μL. The 96-well PCR plates were sealed with optical seal, shaken, and centrifuged after the protein and the molecule were added. Thermal scanning (25 to 80 °C at 0.5 °C/min) was performed using a Bio-Rad 96 Well Real-Time PCR System. Fluorescence signals of all samples have been normalized to relative values of 0% (the lowest fluorescence signal) and 100% (the highest fluorescence signal), respectively. A negative control lacking molecule was present on each assay plate. All assays were performed in triplicate.

### 2.9. Isothermal Titration Calorimetry (ITC)

ITC was carried out according to a previous study with some modifications [40] using ITC200 (Malvern) at 25 °C with a constant stirring rate at 750 rpm. GltB was diluted in buffer (50 mM Tris-HCl, PH 7.5, 300 mM NaCl) and glucose was dissolved in the same buffer. The GltB protein solution (370 μL at 17 μM) was prepared, and then degassed and injected into the sample cell, followed by the titration of glucose solution (220 μL at 200 μM). The ligand was injected twenty times (1 μL for injection 1 and 2 μL for injections 2-20), with 120 s intervals between injections. Raw titration data were concentration-normalized and corrected for dilution effects prior to analysis using the “One-binding site model” of the MicroCal version of ORIGIN. The parameters Δ*H* (reaction enthalpy), *K*_A_ (binding constant, *K*_A_ = 1/*K*_D_) and n (reaction stoichiometry) were determined from the curve fit. The changes in free energy (Δ*G*) and entropy (Δ*S*) were calculated from the values of *K*_A_ and Δ*H* with the equation: Δ*G* = −*RT* ln *K*_A_ = Δ*H* − *T*Δ*S*, where R is the universal molar gas constant and T is the absolute temperature.

### 2.10. Phosphorylation of GltR and its Variants by Acetyl Phosphate

Phosphorylation of 6His-GltR by acetyl phosphate was detected with the Pro-Q Diamond phosphorylation gel stain as described by the manufacturer (Invitrogen). Purified 6His-GltR (~2 μg) or 6His-GltR^D56A^ (~2 μg) was added to phosphorylation reaction buffer (10 mM Tris pH 8.0, 5 mM MgCl_2_, 10 mM KCl, 1 mM DTT) supplemented with or without 50 mM acetyl phosphate, and the mixtures were incubated at 37 °C for 30 min. A total of 10 μL aliquots were removed following the addition of 2 μL 5 × SDS loading buffer (50 mM Tris-HCl, pH 6.8, 2% SDS, 0.1% bromophenol blue, 1% mercaptoethanol; 10% glycerol, 100 mM DTT). The samples were resolved on a 12% SDS polyacrylamide gel at 4 °C, and then the gel was immersed in the fixing solution (10% acetic acid, 50% methanol) for 60 min and then washed three times with deionized water each for 10 min. Subsequently, the gel was stained with Pro-Q Diamond phosphoprotein gel stain for 60~90 min, followed by washing twice with deionized water each for 5 min. Fluorescent output was recorded using Tanon-5200 multi, according to the manufacturer’s recommendation.

### 2.11. Chromatin Immunoprecipitation (CHIP) and CHIP-seq Data Analysis

For generating the plasmid mini-*gltR-flag-lacZ*, primers pAK1900-mini-F (with *Xba*I site) and pAK1900-mini-R (with *Xba*I site) were used. Briefly, a ~1 kb PCR product covering the *lac* promoter of pAK1900 plasmid and the *gltR*-*flag* was amplified from p-*gltR-flag* plasmid (Appendix A) DNA. The PCR product was cloned into integrated mini-CTX-*lacZ* vector [34].

Chromatin immunoprecipitation (CHIP) was performed as previously described [41] with minor changes. The experiments were performed with three biological replicates. Wild type *P. aeruginosa* PAO1 containing mini-*gltR-flag-lacZ* (Appendix A) was cultured in M8 minimal medium supplemented with 10 mM glucose overnight at 37 °C with shaking (250 rpm), diluted 10-fold in fresh M8 minimal medium supplemented with 10 mM glucose. The diluted cultures were grown in a 1 L Erlenmeyer flask with a flask volume-to-medium ratio of 5:1 at 37 °C with shaking (250 rpm). After incubated at 37 °C for 6 h (OD_600_ ≈ 0.5), cultures were treated with 1% formaldehyde for 10 min at 37 °C with shaking (100 rpm). Cross-linking was stopped by addition of glycine to a final concentration of 125 mM. The 75 mL cultures were collected by centrifugation, then pellets were washed twice with a Tris buffer (20 mM Tris-HCl pH 7.5, 150 mM NaCl), and then resuspended in 500 μL IP buffer [50 mM HEPES-KOH pH 7.5, 150 mM NaCl, 1 mM EDTA, 1% Triton X-100, 0.1% SDS, protease inhibitor cocktail (Thermo, Cat#:78420)] and the DNA was sonicated (Scientz, JY92-IIDN) to shear DNA to an average size of 200–500 bp (10% total output, 1-second on, 2-second off, for 60 min on ice). Insoluble cellular debris was removed by centrifugation (12,000 rpm, 4 °C) for 5 min and the supernatant was saved. Immunoprecipitation samples were incubated overnight at 4 °C on a rotating wheel with protein A beads (Smart-Lifesciences, Cat#: SA015C), which was incubated with 100 μL agarose-conjugated anti-flag antibodies (Cat#: AGM12165, Aogma) in IP buffer. Input samples without antibody were set up as negative control. Washing, crosslink reversal, and purification of the CHIP DNA were conducted by following previously published protocols [42] with minor changes. Briefly, the beads were then collected and washed once with 1 mL of IP buffer, once with 1 mL of IP buffer plus 500 mM NaCl, once with 1 mL of IP buffer plus 250 mM LiCl, and once with 1 mL of Tris-EDTA buffer (pH 7.5). Immunoprecipitated complexes were eluted from the beads by treatment with 100 μL elution buffer (50 mM Tris-HCl pH 7.5, 10 mM EDTA, 1% SDS) at 65 °C for 20 min. Samples were then treated with Protease K (TIANGEN) and cross-links were reversed by incubation overnight at 65 °C. Overnight samples were then treated with 50 μg RNase A at 37 °C for 1 h. DNA was extracted twice with phenol-chloroform, precipitated with ethanol at −80 °C for 4 h, and then resuspended in 20 μL of deionized water.

DNA fragments (200 to 500 bp) were selected for library construction, and sequencing libraries were prepared using the NEBNext Ultra II DNA Library Prep kit. The final DNA libraries were validated with Agilent 2100 Bioanalyzer using Agilent High Sensitivity DNA Kit, then the libraries were sequenced using the HiSeq PE150 system (Illumina) completed by Sangon Biotech (Shanghai, China). CHIP-seq reads were mapped to the *P. aeruginosa* genomes, using BBMap with two mismatches allowed. Only the uniquely mapped reads were kept for the subsequent analyses. The enriched peaks were identified using MACS14 software (version 1.4.2) [43] and were visualized and screenshots prepared using Integrative Genome Viewer (IGV) (Broad Institute, version 2.4.6). We filtered peaks called by MACS14 by requiring an adjusted score (i.e., −log10 *p*-value) of at least 50 in order to ensure that we had a high quality peak annotation, and peaks with a fold enrichment lower than 3-fold changes were also filtered out. A total of 16 peaks (≥10-fold enrichment, in average) were used to define the GltR-binding motif using the MEME tool [44]. Annotation of the peaks was performed using the SnpEff (4.3s) (http://www.htslib.org/, accessed on 1 May 2019). The CHIP-seq data files have been deposited in NCBI’s (National Center of Biotechnology Information) Gene Expression Omnibus (GEO) and can be accessed through GEO Series accession number GSE153848, with the following BioSample accession numbers: SAMN15457028 to SAMN15457031.

### 2.12. Electrophoretic Mobility Shift Assay (EMSA)

EMSA experiments were performed as previously described [38] with some modifications. Briefly, reactions were conducted in a mixture containing DNA probe (40 ng), various amounts of purified 6His-GltR or 6His-GltR^D56A^, followed by the addition of binding buffer (10 mM Tris-Cl, pH 7.8, 1 mM DTT, 10% glycerol, 5 mM MgCl_2_, 10 mM KCl) up to 15 μL. The mixtures were incubated at 37 °C for 30 min. When indicated, 50 mM acetyl phosphate was added to the solution. Native polyacrylamide gel (6%) was run in 0.5 × TBE buffer at 85 V at 4 °C. The gel was stained with GelRed nucleic acid staining solution (Biotium) for 5 min, and then the DNA bands were visualized by gel exposure to 260 nm UV light.

DNA probes were amplified by PCR using *P. aeruginosa* PAO1 genomic DNA as templates, and the set of primer pairs indicated in Appendix A. All PCR products were purified by using a QIAquick gel purification kit (QIAGEN). For *gltB-p,* a 302 bp DNA fragment covering the promoter region of *gltB* (from nucleotide −266 to nucleotide +36 relative to the start codon of *gltB*) was amplified with primer pair *gltB*-F (EMSA)/*gltB*-R (EMSA). For *opgG-p*, a 513 bp DNA fragment covering the promoter region of *opgG* (from nucleotide −493 to nucleotide +20 relative to the start codon of *opgG*) was amplified with primer pair *opgG*-F (EMSA)/*opgG*-R (EMSA). For *ctpH-p*, a 328 bp DNA fragment covering the promoter region of *ctpH* (from nucleotide −229 to nucleotide +99 relative to the start codon of *ctpH*) was amplified with primer pair *ctpH*-F (EMSA)/*ctpH*-R (EMSA). For the control DNA fragment used in the EMSAs, *exsA-p* is a 376 bp DNA fragment amplified from the promoter region *exsA* (from nucleotide −343 to nucleotide +33 relative to the start codon of *exsA*) with primer pair *exsA*-F (EMSA)/*exsA*-R (EMSA).

### 2.13. Dye Primer-Based DNase I Foot-Printing Assay

The published DNase I footprint protocol was modified [45]. Briefly, reactions were conducted in a mixture containing 6-carboxyfluorescein(6-FAM)-labeled promoter DNA (300 ng), indicated concentration of purified 6His-GltR, followed by the addition of binding buffer (10 mM Tris-Cl, pH 7.8, 1 mM DTT, 10% glycerol, 5 mM MgCl_2_, 10 mM KCl, 50 mM acetyl phosphate) up to 50 μL. The mixtures were incubated at 37 °C for 30 min. 0.05 unit of DNase I (Promega Biotech Co., Ltd, Cat#:137017) was added to the reaction mixture and incubated for 1~10 min. The digestion was terminated by adding 200 μL phenol-choroform-isoamyl (25:24:1) and 145 μL deionized water. The digested DNA fragments were isolated by ethanol precipitation. A total of 5 μL of digested DNA was mixed with 4.9 μL of HiDi formamide and 0.1 μL of GeneScan-500 LIZ size standards (Applied Biosystems). The sample was detected by a 3730XL DNA analyzer, and the result was analyzed with GeneMapper software (Applied Biosystems). The dye primer based Thermo SequenaseTM Dye Primer Manual Cycle Sequencing Kit (Thermo, Lot:4313199) was used in order to more precisely determine the sequences of the GltR-protection region after the capillary electrophoresis results of the reactions were aligned, and the corresponding label-free promoter DNA fragment was used as template for DNA sequencing. Electropherograms were then analyzed with GeneMarker v1.8 (Applied Biosystems).

DNA probes were amplified by PCR using *P. aeruginosa* PAO1 genomic DNA as templates, and the set of primer pairs indicated in Appendix A. For the DNase I footprinting assay of *gltB*, a 302 bp FAM (carboxyfluorescein) labeled promoter DNA (nucleotide −266 to nucleotide +36 relative to the start codon of *gltB*) was generated using the primer pair *gltB*-F (EMSA)-FAM/*gltB*-R (EMSA). For the DNase I footprinting assay of *opgG*, a 513 bp FAM (carboxyfluorescein) labeled promoter DNA (nucleotide −493 to nucleotide +20 relative to the start codon of *opgG*) was generated using the primer pair *opgG*-F (EMSA)-FAM/*opgG*-R (EMSA). For the DNase I footprinting assay of *ctpH*, a 328 bp FAM (carboxyfluorescein) labeled promoter DNA (nucleotide −229 to nucleotide +99 relative to the start codon of *ctpH*) was generated using the primer pair *ctpH*-F (EMSA)-FAM/*ctpH*-R (EMSA). PCR products were purified by with QIA quick gel purification kit (QIAGEN).

### 2.14. Western Blotting

Strains were grown in M8 minimal medium in the presence or absence of 10 mM glucose at 37 °C overnight with shaking at 250 rpm, diluted 25-fold in fresh M8 minimal medium supplemented with or without 10 mM glucose. The diluted cultures were grown in a 20 mL tube with a tube volume-to-medium volume ratio of 5:1, shaking with 250 rpm at 37 °C for about 6 h (OD_600_ ≈ 0.4). Cell cultures were harvested to OD_600_ 0.8 by centrifugation at 14,000 rpm for 15 min and resuspended in 80 μL deionized water and 20 μL 5 × SDS-PAGE loading buffer [50 mM Tris-HCl, pH 6.8, 2% SDS, 0.1% bromophenol blue, 1% mercaptoethanol, 10% glycerol, 100 mM dithiothreitol (DTT)] and then heated at 100 °C for 10 min.

A total of 10 μL of loading buffer-treated sample as described above was loaded onto a 12% SDS polyacrylamide gel. SDS-PAGE was carried out at 80 V for 15 min followed by 120 V for 80 min. Samples resolved on gels were transferred to PVDF (Bio-Rad) membranes through semi-dry transfer assembly (Bio-Rad) for 28 min at room temperature. The membrane was incubated with the primary antibody in 5% (wt/vol) skim milk at 4 °C overnight following the blocking step (15 mL of 5% (wt/vol) skim milk) at room temperature for 2 h, and then washed four times at room temperature each for 30 min in TBST buffer (10 mM Tris-HCl, pH 7.5, 150 mM NaCl, and 0.1% Tween 20). Then, membranes were incubated with the secondary antibody for 2 h at room temperature and washed four times each for 30 min in TBST. The chemiluminescent detection reaction was performed and detected by Tanon-5200 multi, according to the manufacturer’s recommendation.

GltB proteins were detected by using a rabbit anti-GltB polyclonal antibody (prepared by immunizing a rabbit with a N-terminal 6His-tagged full-length GltB protein, Sangon Biotech Co., Ltd.) at 1:2000 followed by a secondary, an anti-rabbit IgG antibody conjugated to horseradish peroxidase (HRP) (Code#: NA934, GE Healthcare) at 1:5000. GtrS-YFP proteins were detected by using a rabbit anti-GFP antibody (KleanAB, #P100009) at 1:2000 followed by a secondary, sheep anti-rabbit IgG antibody conjugated to horseradish peroxidase (HRP) (Code#: NA934, GE Healthcare) at 1:5000. GltR-Flag proteins were detected by using a mouse anti-Flag monoclonal antibody (Cat#: AGM12165, Aogma) at 1:2000 followed by a secondary, sheep anti-mouse IgG antibody conjugated to horseradish peroxidase (HRP) (Code#: NA931, GE Healthcare) at 1:5000. For detection of RNAP protein, anti-RNAP (Neoclone, #WP003) antibody at 1:2000 and anti-mouse IgG antibody conjugated to horseradish peroxidase (HRP) (Code#: NA931, GE Healthcare) at 1:5000 were used.

### 2.15. Localized Surface Plasmon Resonance

The interaction of 6His-GltB and 6His-GtrS^LBD^ was analyzed using an OpenSPR localized surface plasmon resonance (Nicoya Lifesciences, Waterloo, Canada). 6His-GtrS^LBD^ protein was fixed on the COOH sensor chip by capture-coupling, then 6His-GltB at concentrations of 1000 nM, 500 nM, 250 nM, 125 nM were injected sequentially into the chamber in running buffer (50 mM Tris, 50 mM NaCl, pH 7.0) at 25 °C. The binding time and disassociation time were both 375 s, the flowrate was 20 μL/min, the chip was regenerated with 0.02% SDS. A one to one diffusion corrected model was fitted to the wavelength shifts corresponding to the varied drug concentration. The data were retrieved and analyzed with TraceDrawer software (Ridgeview Instruments ab, Sweden). Kinetic parameters were calculated using a global analysis, and the data was fitted to a one to one model.

The similar experimental procedure was used to analyze the interaction of 6His-GltB with 6His-PctA^LBD^. 6His-GltB protein was fixed on the COOH sensor chip by capture-coupling, then 6His-PctA^LBD^ at concentrations of 1500 nM, 1200 nM, 750 nM, 500 nM were injected sequentially into the chamber in running buffer. The binding time and disassociation time were both 528 s.

For the analysis of the interaction of 6His-GltB with 6His-PA2788^LBD^, 6His-GltB protein was fixed on the COOH sensor chip by capture-coupling, then 6His-PA2788^LBD^ at concentrations of 500 nM, 400 nM, 200 nM, 100 nM were injected sequentially into the chamber in running buffer. The binding time and disassociation time were both 475 s. The methods used were otherwise identical to those described above.

### 2.16. Bacterial Two-Hybrid Analysis

For monitoring of protein-protein interaction in vivo, the Euromedex bacterial two-hybrid (BACTH, for “Bacterial Adenylate Cyclase-based Two-Hybrid”) system (Cat No:EUK001) was used [46,47]. BACTH relies on reconstitution of activity of the split *Bordetella pertussis* adenylate cyclase toxin. Reconstitution and thus cAMP production occurs through interaction of candidate proteins fused to the separately encoded T18- and T25-fragments of the *B. pertussis* toxin. The plasmid-encoded fusion genes are tested in *E. coli* strain BTH101, which lacks endogenous adenylate cyclase activity. Interaction can be monitored quantitatively by measuring activity of β-galactosidase, whose synthesis depends on cAMP-CRP. Plasmids pKT25 and pUT18C were used for construction of in-frame fusion of the candidate genes to the 3′ ends of the sequences encoding T25 and T18, respectively.

The BACTH plasmid encoding the T18-GltB fusion protein was obtained by amplification of the *gltB* gene using primers *gltB*-pUT-F (with *Xba*I site) and *gltB*-pUT-R (with *Kpn*I site) indicated in Appendix A and subsequent insertion of the PCR-fragment between the *Xba*I and *Kpn*I sites on the plasmid pUT18C. Note that within the T18-GltB fusion gene, *gltB* lacks the first 25 codons encoding the N-terminal signal sequence (https://www.uniprot.org/, accessed on 1 May 2018). Therefore, the GltB sequence fused to T18 corresponds to the mature form of GltB as present in the periplasm. For construction of BACTH plasmid T25-GtrS^LBD^, the codons 29–199 of *gtrS* were amplified by PCR of the *gtrS* gene using the primer pair *gtrS*-LBD-pKT-F/*gtrS*-LBD-pKT-R (*Xba*I/*Kpn*I sites) indicated and inserted between *Xba*I and *Kpn*I sites on the plasmid pKT25. All constructs were sequenced to ensure that no unwanted mutations resulted. BTH101 was co-transformed with the plasmid carrying the T18 and T25 fusion genes.

Transformants were selected using kanamycin (50 µg/mL) and ampicillin (100 µg/mL) on LB agar plates, and were subsequently inoculated onto the M63/maltose agar plates supplemented with kanamycin (25 µg/mL), ampicillin (50 µg/mL), X-Gal (40 µg/mL), and IPTG (0.5 mM), incubated at 30 °C for up to 6 days. Positive interactions between the target proteins were indicated by the appearance of blue colonies. Protein-protein interaction was also detected by measuring the β-galactosidase activities in bacteria grown in LB broth containing kanamycin (50 µg/mL), ampicillin (100 µg/mL), and 0.5 mM IPTG at 30 °C for 12 h with shaking at 250 rpm, according to the protocol of BACTH system.

### 2.17. Co-Immunoprecipitation Coupled with Mass Spectrometric (CoIP-MS)

*P. aeruginosa* strains [PAO1/p-*gtrS*-YFP and Δ*gltB*/p-*gtrS*-YFP (negative control)] were grown in 20 mL of LB broth containing carbenicillin (150 µg/mL) at 37 °C overnight with shaking at 250 rpm. Overnight cultures were washed twice in 10 mL of fresh M8 minimal medium and suspended in 20 mL of fresh M8 minimal medium supplemented with 10 mM glucose, then shaking (250 rpm) at 37 °C for 1 min. The 20 mL cultures were collected by centrifugation (7800 rpm) at 4 °C for 5 min, washed once in PBS buffer, and then suspended in 1 mL of lysis buffer provided in the Co-IP kit (Pierce) supplied with 1 μL protease inhibitor cocktail (Thermo, Cat#:78420). The mixture was homogenized by mechanical disruption (Fast Prep FP2400 instrument; Qbiogene) and then the debris was removed by centrifuging at 2300 g for 9 min and filtrating through 0.44 μm filters. The 700 µL clarified lysates were incubated with the resins immobilized with a specific antibody (a rabbit anti-GltB polyclonal antibody) overnight at 4 °C. The bound materials were washed and eluted following the manufacturer’s recommendation (Thermo, Cat#: 26149). The immunocomplexes eluted from the resins by western blot with specific antibodies, as indicated.

Co-IP samples as described above were evaporated by lyophilizer (ThermoFisher, SAVANT SPD111V), then resuspended in 16 μL distilled water and mixed with 4 μL 5 × SDS-PAGE loading buffer and loaded onto a 12% polyacrylamide gel. SDS-PAGE was carried out at 80 V for 15 min followed by 120 V for 80 min. The gel was first visualized with Coomassie brilliant blue stain, and then each gel lane was divided into small fractions and digested with trypsin before mass spectrometric analysis. For trypsin digestion, the protein gel was reduced with 20 mM DTT (50 mM NH_4_HCO_3_) at 56 °C for 30 min and alkylated with iodoacetamide (50 mM NH_4_HCO_3_) at a final concentration of 100 mM at room temperature for 20 min in darkness. Then proteins were digested overnight with 1:50 (protein: enzyme, *w*/*w*) trypsin (Promega) at 37 °C.

The proteome analysis was performed on an Orbitrap Q-Exactive HF (Thermo Fisher Scientific) platform connected to an online nanoflow EASY nLC1200 HPLC system (Thermo Fisher Scientific). The tryptic peptides were loaded onto 15 cm columns packed in-house with C18 1.9 μm ReproSil particles (Dr. Maisch GmbH) and separated with a 60-min gradient at a flow rate of 300 nL/min. A homemade column oven maintained the column temperature at 50 °C. A data-dependent acquisition MS method was used, in which one full scan (350–1500 m/z, R = 70,000 at 200 m/z) at a target of 3e6 ions was first performed, followed by top 15 data-dependent MS/MS scans with higher-energy collisional dissociation (HCD) at a resolution of 17,500 at 200 *m*/*z*. Other instrument parameters were set as follows: 27% normalized collision energy (NCE), 1e3 AGC target, 100 ms maximum injection time, 2.0 *m*/*z* isolation window. Raw mass spectrometry data were processed using MaxQuant version 1.6.0.1 [48] against the *P. aeruginosa* database, with a false-discovery rate (FDR) < 0.01 at the level of proteins and peptides. Carbamidomethyl (C) was selected as a fixed modification, oxidized methionine (M) and acetylation (protein N-term) as variable modifications. Control sample was also analyzed in parallel to distinguish the background proteins. After excluding the background interfering proteins, the rest of proteins were identified as potential GltB interacting protein. The mass spectrometry proteomics data have been deposited to the ProteomeXchange Consortium via the PRIDE partner repository with the dataset identifier PXD020213.

### 2.18. Drosophila Melanogaster Infection Assays

*Drosophila* infection was adapted from the fly needle pricking assay according to the protocol as previously described [49,50]. All *P. aeruginosa* strains were grown in PB medium at 37 °C overnight with shaking at 250 rpm, diluted 100-fold in fresh PB medium. Subsequently, the diluted cultures were grown in a 20-mL tube with a tube volume-to-medium ratio of 5:1 at 37 °C with shaking (250 rpm). After incubation for 3 h, bacteria were harvested by centrifugation, washed twice and suspended in PBS buffer (NaCl 8 g, KCl 0.2 g, Na_2_HPO_4_ 1.44 g, KH_2_PO_4_ 0.24 g, per liter) and then diluted the OD_600_ to 0.8. The dorsal side of the thorax of carbon dioxide-anesthetized male flies (~5 days old) was pricked with a sterilized tungsten needle dipped in the appropriate bacterial suspension or PBS buffer. The pricked flies were grown in the fly-food vial, kept at 25 °C and 65% humidity. Fly survival was counted and survival curves were processed with GraphPad Prism (GraphPad Software, Inc., San Diego, CA, USA) to perform a statistical log-rank (Mantel-Cox).

### 2.19. Mouse Model of Acute Pneumonia

*P. aeruginosa* strains were grown in PB medium at 37 °C overnight with shaking (250 rpm), diluted 100-fold in fresh PB medium. The diluted cultures were grown in a 100 mL Erlenmeyer flask with a flask volume-to-medium ratio of 5:1 at 37 °C with shaking (250 rpm). After incubated at 37 °C for 3 h until the cultures reached OD_600_ 0.8. Bacteria were collected by centrifugation, washed and suspended in PBS buffer. Mouse infections were carried out as described previously [38], using 8-week-old female C57BL/6J mice. Mice were anaesthetized with pentobarbital sodium (intraperitoneal injection) and intranasally infected with 4 × 10^8^ CFU of each bacterial isolate. After that, animals were sacrificed 18 h post infection. Lungs were aseptically removed and homogenized in PBS plus 0.1% Triton X-100 in order to obtain single-cell suspensions. Serial dilutions of each organ were plated on *Pseudomonas* isolation agar (PIA) plates. Bacterial burden per organ was calculated and is expressed as a ratio of the inoculum delivered per animal. Statistical analysis (the Mann–Whitney test) was performed using GraphPad Prism.

## 3. Results

### 3.1. GtrS-GltR Is Required for Glucose Utilization in Gcd-Deficient P. aeruginosa Strain

To examine the role of GtrS-GltR TCS in glucose utilization, we deleted the *gtrS-gltR* locus encoding the GtrS-GltR TCS in the *P. aeruginosa* reference strain PAO1, yielding Δ*gtrS-gltR*, and monitored bacterial growth in a chemically defined minimal medium (MM) amended with glucose as the sole carbon source. The Δ*gtrS-gltR* mutant grew with a similar kinetic to the wild type (WT) PAO1 strain (Figure 1B), indicating that the GtrS-GltR TCS is not essential for glucose utilization by *P. aeruginosa* PAO1.

Like *gtrS-gltR* deletion, the loss of *gcd* encoding membrane-bound glucose dehydrogenase (Gcd), which catalyzes the periplasmic oxidation of glucose (Figure 1A), has no obvious effect on the ability of *P. aeruginosa* PAO1 to use glucose as its sole source of carbon for growth (Figure 1B). This observation suggests that the oxidative pathway of glucose degradation (Figure 1A) is not required for glucose utilization by *P. aeruginosa* PAO1 as well. However, the growth of Δ*gcd*Δ*gtrS-gltR* triple mutant strain lacking both *gcd* and *gtrS-gltR* on glucose used as the sole carbon source was abolished (Figure 1B), although growth on either gluconate or 2-KG was sustained (Appendix A). The growth defect was restored when the Δ*gcd*Δ*gtrS-gltR* was transformed with a plasmid that expressed either *gtrS-gltR* or *gcd* (Figure 1B), indicating that GtrS-GltR TCS modulates glucose utilization independently of Gcd, and vice versa. These results suggest that GtrS-GltR is essential for glucose utilization in *P. aeruginosa* strain when Gcd is absent, which also support the notion that in aerobically grown *Pseudomonas* species extracellular glucose can be catabolized by either the phosphorylative pathway or the oxidative pathway (Figure 1A).

### 3.2. GtrS-GltR Positively Regulates the gltBFGK-oprB Operon

Because *gltR* and *gltB*, the first gene of the *gltBFGK-oprB* operon involved in glucose uptake, are both essential for glucose transport activity in *P. aeruginosa* [27,53], we reasoned that GtrS-GltR TCS may activate *gltBFGK-oprB* operon in response to glucose. Using promoter-fusion analysis, we found that in the absence of glucose the promoter activity of *gltB* (i.e., *gltB-lux*) was low, and it was not obviously affected by the deletion of *gtrS-gltR* locus (- Glucose in Figure 2A). In the presence of glucose, the expression level of *gltB-lux* in WT PAO1 was approximately 64.5-fold higher than in the absence of glucose (Figure 2A). However, the addition of glucose failed to increase the expression of *gltB-lux* in the *gtrS-gltR* deletion mutant (Δ*gtrS-gltR*), although it did this for the complemented strain (Δ*gtrS-gltR*/p-*gtrS-gltR*) (Figure 2A). Moreover, we found that deletion of *gtrS*, which encodes the SK of the GtrS-GltR TCS, completely abolished the glucose-induced *gltB-lux* (Figure 2A). These data suggest that GtrS-GltR TCS is required for the transcription of *gltBFGK-oprB* operon in response to glucose. Furthermore, we observed that glucose failed to increase the expression of *gltB-lux* in the Δ*gtrS-gltR* mutant complemented with a plasmid carrying the functional *gtrS* (i.e., p-*gtrS*) (Figure 2A), which indicates that *gltR* is required for the glucose-induced expression of *gltB-lux*.

Like some other RRs [38,56], the N-terminally 6His-tagged GltR protein (i.e., 6His-GltR) can be phosphorylated by the low molecular weight phosphodonor acetyl phosphate (Appendix A). The 6His-GltR mutant (i.e., 6His-GltR^D56A^), in which the predicted phosphoryl group-accepting Asp56 was replaced by alanine, lost its capacity to get phosphorylated (Appendix A), supporting that D56 is the target residue for phosphorylation. Using electrophoretic mobility shift assays (EMSAs), we found that the addition of acetyl phosphate increased the ability of 6His-GltR to bind to the *gltB* promoter DNA (Appendix A), while it failed to do this for the 6His-GltR^D56A^ proteins (Appendix A). However, the negative control *exsA* promoter was unbound (Appendix A). These results indicate that phosphorylation of GltR increases its binding to its target DNA.

Using DNase I foot-printing assays, we found that the 6His-GltR protected two adjacent sites (I: −44 to −13, II: −11 to +15; relative to the transcriptional initiation site) of the *gltB* promoter from DNase I digestion (Figure 2B). These results are consistent with a previous study [26], showing that an DNA sequence (5′-GTGAAAAACCGGA-3′) at position −22 to −10 relative to the transcriptional initiation site of *gltB* is a GltR-binding site (Figure 2C). However, when compared to those of either the published PAO1 sequence or our laboratory PAO1 reference strain, a single-nucleotide deviation (SNP) and a 1 bp-deletion (5′-GTGAAAAACCGGA-3′ *vs* 5′-GTGACAAACCCGGA-3′) were found in the GltR-binding element of the PAO1 strain used in the previous study by Daddaoua et al. [26] (Figure 2C), and this observation supports the notion that *P. aeruginosa* PAO1 shows an ongoing microevolution [57,58].

Moreover, we found that glucose failed to increase the expression of *gltB-D-lux*, a *gltB-lux* variant lacking part of GltR-binding site (i.e., 5′-GTGACAAA-3′), in WT *P. aeruginosa* PAO1 (Appendix A), suggesting that the binding of GltR to the promoter DNA is essential for the glucose-induced expression of *gltBFGK-oprB* operon. In addition, disruption of the His280 of GtrS, presumably an essential residue for histidine kinase function [49], abolished the function of GtrS in activating *gltB-lux* in response to glucose (Appendix A). Collectively, these results clearly suggest that GtrS-GltR TCS positively regulates the *gltBFGK-oprB* operon via the binding of GltR to the promoter of *gltB*.

### 3.3. 2-KG and 6PG Are Not Required for Glucose-Induced Activation of GtrS-GltR

It has been reported that 2-KG and 6PG bind to the ligand-binding domain of GtrS and increase its autophosphorylation [26]. Intrigued by these findings, we reasoned that 2-KG and 6PG might be factors involved in glucose-induced expression of *gltB*. To test this hypothesis, we examined the effects of 2-KG and 6PG on the expression of *gltB-lux*. The addition of either 2-KG (5 mM) or gluconate (5 mM), which can be phosphorylated to 6PG by gluconokinase (GnuK), increased the expression of *gltB* in WT *P. aeruginosa* PAO1 by approximately 3.5-fold (Figure 3A), a much lesser extent than that of glucose (about 258-fold, at a final 5 mM concentration) (Figure 3A), implying that 2-KG and 6PG play a far lesser role in determining the activation of GtrS-GltR than glucose.

Because glucose can be oxidized in the periplasm by glucose dehydrogenase (Gcd) and gluconate dehydrogenase (Gad) to gluconate and 2-KG (Figure 1A), we examined the role of Gcd and Gad on glucose-induced activation of GtrS-GltR TCS. We measured the expression level of *gltB-lux*, an indicator for the transcriptional regulatory function of GtrS-GltR, in the WT PAO1 strain, the *gcd* deletion mutant (Δ*gcd*), and the *gad* deletion mutant (Δ*gad*), when bacteria were cultured in M9 minimal medium supplemented with or without glucose (5 mM). The glucose-induced expression level of *gltB-lux* was decreased by approximately 1.5-fold upon the deletion of either *gcd* or *gad* (Figure 3B), suggesting that either gluconate or 2-KG has little effect on the activation of GtrS-GltR in response to glucose.

Using promoter fusion assays, we showed that *gcd* deletion, which presumably abolishes the production of gluconate and 2-KG [21,59], severely decreases (more than 70% reduction) the promoter activities of *gntR* and *ptxS* (Figure 3C,D). These results are well in line with previous studies showing that gluconate and 2-KG, respectively, induces the expression of *gntR* [25] and *ptxS* [52], and support the proposed glucose oxidative pathway in the periplasm of *P. aeruginosa* (Figure 1A). Moreover, we observed that exogenously applied glucose, even at low micromolar concentrations (i.e., 10 µM), was capable of inducing the expression of *gltB-lux* in *P. aeruginosa* PAO1 but failed to do for the expression of either *gntR-lux* or *ptxS-lux* (Figure 3E). Based on these data, we concluded that unidentified mechanisms for glucose-mediated activation of GtrS-GltR exists in *P. aeruginosa*.

### 3.4. Genome-Wide Mutagenesis Identifies gltB Is Crucial for GtrS-GltR Activation

To explore the mechanism by which glucose activates GtrS-GltR, we performed a screen using a library of *P. aeruginosa* PAO1 transposon mutants and a promoter fusion analysis with *lacZ* reporter (i.e., *gltB-lacZ*), an indicator of the GtrS-GltR transcriptional activity (Figure 2A). From a transposon mutant library with appropriately 50,000 mutants of WT PAO1, we identified a total of 24 colorless colonies (Appendix A, Figure 4A). Of them, 13 had a transposon insertion in either the *glk-gtrS-gltR* operon [60] or its promoter region, and the other 11 had a transposon insertion in the promoter or the coding region of *gltB* (Appendix A, Figure 4A).

To verify the role of *gltB* in the transcriptional regulatory activity of GtrS-GltR, we deleted *gltB* in *P. aeruginosa* PAO1, yielding Δ*gltB* mutant. Similar to the original transposon mutant, Δ*gltB* mutant carrying the *gltB-lacZ* reporter showed colorless colonies when bacteria were grown on M8 minimal agar medium supplemented with β-galactosidase (X-gal) (Appendix A). Ectopic expression of *gltB* in the Δ*gltB* mutant forms intense blue colonies (Appendix A), indicative of increased expression of *gltB-lacZ*. These results suggest that GltB plays an important role in the transcriptional regulatory activity of GtrS-GltR against the *gltBFGK-oprB* operon. Similar result was obtained when the *gltB-lux* was used as an indicator for the transcriptional regulatory activity of GtrS-GltR (Figure 4B). Moreover, we found that *gltB* deletion decreased the glucose-induced promoter activity (Figure 4C–E) of several known GltR-regulated genes including *glk*, *edd*, and *toxA* [26], reinforcing the likelihood that that GltB is crucial for the activation of GtrS-GltR in response to glucose.

### 3.5. GltB Binds to a Number of Membrane-Spanning Proteins Including GtrS

Because GltB is a periplasmic protein [53,61,62], and that GtrS is a membrane-bound histidine kinase [26,63], we thus hypothesized that GltB may interact with GtrS. Using co-Immunoprecipitation (Co-IP) experiments with *P. aeruginosa* PAO1 cells expressing yellow fluorescent protein (YFP)-tagged GtrS (GtrS-YFP), we were able to show that GltB directly interacted with GtrS (Figure 5A). To further confirm this observation, the Co-IP samples were subjected to liquid chromatography-tandem mass spectrometry (LC-MS/MS) analysis. As a result, 250 proteins, including GtrS and a number of membrane-bound proteins, such as PctA, PctB, PilJ, and PA2788 [64,65], were identified as potential GltB interacting proteins (Figure 5B, Appendix A).

Using surface plasmon resonance (SPR) analysis, we determined that 6His-GltB bound to 6His-GtrS^LBD^ (N-terminally 6His-tagged ligand-binding domain of GtrS, residues 29–199) with a dissociation constant (*K*_D_) of 0.1 μM (Figure 5C), which is much lower than the dissociation constant of either 2-KG (i.e., 5 μM) or 6-phosphogluconate (i.e., 98 μM) binding to the recombinant GtrS ligand-binding domain [26]. We also found that 6His-GltB bound to 6His-PctA^LBD^ (N-terminally 6His-tagged ligand-binding domain of PctA, residues 30–278) and 6His-PA2788^LBD^ (N-terminally 6His-tagged ligand-binding domain of PA2788, residues 44–179) with *K*_D_ values of 0.8 μM and 0.4 μM, respectively (Appendix A), indicating that GltB may have broad function in the responses of *P. aeruginosa* to glucose. To further examine the binding of GltB to GtrS, we employed a bacterial two-hybrid system and demonstrated GltB was capable of interacting with GtrS in *Escherichia coli* (Figure 5D,E). Based on these data, we concluded that GltB interacts with GtrS in *P. aeruginosa.*

### 3.6. GltB Requires Its Glucose-Binding Pocket to Activates GtrS-GltR

It has been established that GltB is a glucose-specific binding protein (Figure 1A). In line with this, the addition of glucose increased the melting temperature (*Tm*) of GltB in a concentration-dependent manner (Figure 6A). This result suggests that GltB binds to glucose because when a protein binds to a ligand, its melting temperature (*Tm*) might shift [66,67]. Using isothermal titration calorimetry (ITC) analysis, we found that 6His-GltB bound to glucose with the equilibrium dissociation constant (*K*_D_) of 1.37 μM (Figure 6B), a value close to that measured in either surface plasmon resonance (SPR) [55] or the equilibrium dialysis experiment [53].

Twelve residues (i.e., W35, W36, E41, G68, Q90, K92, W250, W270, N301, D303, K339, and H379) present in the substrate binding pocket of *Pseudomonas putida* CSV86 GltB are directly involved in glucose binding [54,55]. To examine the role of the substrate binding pocket of *P. aeruginosa* GltB in the activation of GtrS-GltR, five residues W33, W34, K90, W268, and D301, which correspond to the W35, W36, K92, W270, and D303 of the *P. putida* CSV86 GltB [54,55], were, respectively, mutated to alanine using site-directed mutagenesis. As shown in Figure 6C, each alanine substitution had no obvious effect on the expression/stability of the GltB protein. However, K90A and W268A amino acid substitutions, respectively, decreased the glucose-induced expression of *gltB-lux* by approximately 50% in a Δ*gltB* strain (Figure 6D,E), and notably either W33A, W34A, or D301A amino acid substitution totally abolished the glucose-induced expression of *gltB-lux* (Figure 6D,E). Additionally, we observed that glucose failed to increase the *Tm* of the GltB^D301A^ (Figure 6F), which suggests that this conserved residue is required for *P. aeruginosa* GltB to bind to glucose. These result suggest that the glucose-binding is required for GltB to exert its regulatory function against the GtrS-GltR TCS.

### 3.7. Genome-Wide Identification of GltR Targets

As aforementioned, GltB binds to GtrS and increases the promoter activity of known GltR-regulated genes (i.e., *gltB*, *glk*, *edd*, and *toxA*) (Figure 4B–E, Figure 5). To further verify the role of GltB in the regulatory function of GtrS-GltR TCS, we sought to explore the additional GltR-regulated genes. To this end, we performed chromatin immunoprecipitation followed by high-throughput sequencing (CHIP-seq) assays with a PAO1::*gltR*-*flag* strain expressing a functional C-terminally tagged GltR protein (GltR-flag). Using the MACS software [43], we identified 55 significantly enriched loci (fold change ≥ 3 and q-value < 0.05) (Appendix A, Figure 7A).

The most enriched locus by CHIP-seq experiments was the promoter region of *gltB* (~75-fold enrichment) (Appendix A, Figure 7A), suggesting that GltR strongly binds to the promoter of *gltB* under the test condition. Using MEME tool [44], we found that the most significant motif (E-value=1.3 × 10^−6^) containing a consensus 5′-GTNACAAA-3′ sequence (in positions 14-21, where N is any nucleotide) (Figure 7B), which matches the GltR-binding site (i.e., GTGACAAA) revealed by footprint assays on the *gltB* promoter (Figure 2B), was enriched in the 101 bp region centered at the summits for all tested 16 peaks (≥10-fold enrichment). These results support the effectiveness of the CHIP-seq procedure and indicate that the determination of GltR-binding consensus sequence was carried out successfully.

Among the 55 potential GltR-binding sites, 61.8% (34 peaks) are in the potential promoters (within 500 bp of the translational start site) of *P. aeruginosa* genes (Appendix A). Because some GltR-binding sites are flanked by divergently transcribed genes (or operons), GltR binding within the 34 regions could potentially control the expression of 112 genes in 48 transcription units (TUs) (Appendix A). Functional analyses showed that these potential GltR-targeted genes are associated with various biological functions including glucose transport (e.g., *gltB*), synthesis of glucans (e.g., *opgGH* operon), phosphate taxis (e.g., *ctpH*), and copper tolerance (e.g., *copR* and *ptrA*) (Appendix A).

### 3.8. GltB Positively Regulates opgGH and ctpH, Two New GltR Targets

As described above, our CHIP-seq experiments showed that *opgGH* operon and *ctpH* are potential GltR targets (Figure 7A,C,D). In *P. aeruginosa*, the *opgGH* operon is required for the biosynthesis of osmoregulated periplasmic glucans (OPGs) involved in osmoprotection, biofilm formation, virulence, and resistance to antibiotics [68], while the *ctpH* encodes a chemotactic transducer for inorganic phosphate [69]. Using EMSA assays, we verified that GltR bound to the promoters of *opgGH* operon and *ctpH* (Appendix A). In the presence of acetyl phosphate, the 6His-GltR bound to those promoter DNAs with stronger affinity, whereas acetyl phosphate has no obvious effect on the binding of 6His-GltR^D56A^ to the promoter DNAs (Appendix A). As the negative control, the *exsA* promoter was unbound (Appendix A). Using DNase I foot-printing experiments, we found that 6His-GltR protected two adjacent sites (−404 to −382; −374 to −359; relative to the start codon) of the *opgG* promoter from DNase I digestion (Figure 7E). Similarly, 6His-GltR also protected two adjacent sites (−180 to −149; −147 to −118; relative to the start codon) of the *ctpH* promoter from DNase I digestion (Figure 7F).

To examine whether GtrS-GltR TCS has a role in modulating the expression of *opgGH* operon and *ctpH*. We measured the promoter activity of *opgG* and *ctpH* in WT PAO1 strain, *gtrS-gltR* deletion mutant (Δ*gtrS-gltR*), and its complemented strain Δ*gtrS-gltR*/p-*gtrS-gltR*, respectively. Deletion of *gtrS-gltR* resulted in an approximately 2-fold decrease in glucose-induced expression of *opgG-lux* in the *P. aeruginosa* PAO1, and the reduced expression of *opgG-lux* was fully restored upon introduction of *gtrS-gltR* with a plasmid (i.e., p-*gtrS-gltR*) (Figure 7G). Similar results were observed when the *ctpH-lux* was examined (Figure 7H). These results, together with the biochemical evidence of direct binding of GltR with the promoter DNA of *opgG* and *ctpH* (Appendix A), suggest that *opgGH* operon and *ctpH* are GltR targets.

We next investigated the role of GltB in the expression of *opgGH* operon and *ctpH*. We performed the promoter analysis again and found that the glucose-induced expression level of both *opgG-lux* and *ctpH-lux* in Δ*gltB* was similar to that of the Δ*gtrS-gltR* (Figure 7G,H). Introduction of plasmid-borne *gltB* into the Δ*gltB* restored the expression of either *opgG-lux* or *ctpH-lux* to wild-type PAO1 levels (Figure 7G,H), demonstrating that like GtrS-GltR TCS, GltB activates the expression of *opgGH* operon and *ctpH* as well. Collectively, these results reinforce the likelihood that GltB is required for the glucose-induced activation of GtrS-GltR (Figure 4).

### 3.9. GltB Is Important for the Pathogenicity of P. aeruginosa

GtrS is important for *P. aeruginosa* infection [60,63]. In line with this, we found that Δ*gtrS* mutant exhibited reduced virulence against infected flies compared with the WT *P. aeruginosa* PAO1 in a *Drosophila melanogaster* nicking infection model (Figure 8A). We also found that the virulence phenotype of Δ*gltB* resembled that of the Δ*gtrS*, and the virulence defect phenotype of Δ*gltB* was complemented in trans with plasmid-borne copies of the *gltB* (Figure 8A).

To further examine the role of GltB and GtrS in the pathogenicity of *P. aeruginosa*, the WT PAO1, Δ*gtrS*, Δ*gltB*, and complemented strains Δ*gtrS*/p-*gtrS* and Δ*gltB*/p-*gltB* were inoculated in a mouse model of acute pneumonia. Figure 8B shows the colony-forming unit (CFU) of bacteria recovered from the lungs compared to the initial inoculum at 18 h post infection, with a geometric mean indicated for each group. The WT PAO1 was able to colonize the lungs of infected mice, and the number increased by approximately 570% over the course of the 18 h infection (Figure 8B). In contrast, the Δ*gtrS* and Δ*gltB* bacteria were, respectively, recovered in numbers approximately at 52% and 38% of initial inoculum dose from lungs (Figure 8B). In complemented strains, the colonization was completely restored to WT levels (Figure 8B). Taken together, these results suggest that GltB is important for the activation of GtrS-GltR TCS during infections.

## 4. Discussion

In this study, we show that periplasmic glucose-binding protein GltB is required for the activation of GtrS-GltR TCS in *P. aeruginosa*. We provide the first proteome-wide analysis of GltB-binding proteins (Figure 5B, Appendix A,) and the first genome-wide analysis of the GltR targets (Figure 7A, Appendix A), laying the ground for a better understanding of signaling pathways engaged by GltB and GtrS-GltR.

GltB is one of periplasmic substrate-binding protein (PBPs) [20,53]. In gram-negative bacteria, PBPs constitute a large family of receptors that recognize and deliver small molecules or ions into the cytoplasm via the cognate inner membrane ATP binding cassette (ABC) transport systems [70]. Although the main function of PBPs is the involvement in transport processes, PBPs were found to possess a number of additional functions and there is increasing evidence for PBP mediated activation of chemoreceptors and sensor kinases [46,71,72,73,74]. In this work we showed that GltB binds to and induces the transcriptional regulatory activity of GtrS-GltR TCS (Figure 4B–E, Figure 5). Interestingly, GltB was also capable of binding to a number of proteins including chemoreceptor PctA [75] and predicted membrane chemotaxis transducer PA2788 [65] (Appendix A, Appendix A,). Therefore, GltB may have a much more profound effect on the cellular function of *P. aeruginosa* than we previously thought, and further studies are required to address this important issue.

We showed that GltR is a positive transcriptional regulator for *gltBFGK* (*gtsABCD*)-*oprB* operon that encodes the glucose transport system [28] (Figure 2A). This result is consistent with some previous studies, where it has been found that GltR acts as an activator of the glucose transport system in *P. aeruginosa* [20,27] and in *Pseudomonas putida* [29]. However, there has also been contradicting report on the role or mechanism of this regulation. Daddaoua and colleagues described GltR as a repressor [26]. These differences may be explained by strain-to-strain differences because *P. aeruginosa* PAO1 shows an ongoing microevolution of genotype and phenotype [57,58]. Indeed, analysis of the promoter sequence of *gltB* pointed to the existence of single-nucleotide polymorphism (SNP) and small deletion in the PAO1 strain used in the studies by Daddaoua et al. [26] compared to the published PAO1 sequence (Figure 2C).

Many studies have suggested that the effect of glucose on *P. aeruginosa* virulence is of medical relevance [11,60,76,77,78,79,80,81,82]. Interestingly, glucose appears to contribute to the virulence evolution of *P. aeruginosa* DK2, a transmissible clone isolated from chronically infected Danish CF patients over a period of 38 years [83,84], by activating BfmRS TCS via the GltB and GtrS-GltR [49]. In addition to acting as a nutrient for *P. aeruginosa* growth, glucose may also serve as a signal molecule that induces virulence functions because micromolar glucose could increase the transcriptional regulatory activity of GtrS-GltR that play an important role in *P. aeruginosa* virulence [63] (Figure 3E, Figure 4E, and Figure 8).

In conclusion, the results from this study suggested that GltB interacts with GtrS and initiates the GtrS-GltR signaling cascade that allows *P. aeruginosa* to respond to the presence of glucose. A deeper understanding of mechanism of action of GltB/GtrS-GltR regulatory axis may lead to more effective methods of treatment of *P. aeruginosa* infections.

## Figures and Tables

**Figure 1 microorganisms-09-00447-f001:**
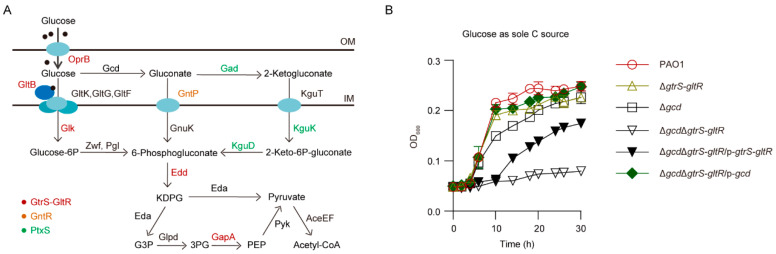
Proposal pathways of glucose metabolism and bacterial growth assays. (**A**) Schematic view of predicted glucose utilization and metabolism pathways in *P. aeruginosa* inferred from previous studies [23,51]. Proteins are shown in different colors according to the regulators (e.g., GtrS-GltR, GntR, and PtxS) that participate in their transcriptional regulation [25,26,52]. OM, outer membrane; IM, inner membrane; OprB, glucose porin; GltB, periplasmic glucose-binding protein [53,54,55]; GltK, ATP-binding component of ABC transporter; GltG and GltF, permease of ABC sugar transporter; GntP, gluconate permease; KguT, 2-ketogluconate (2-KG) transporter; Gcd, glucose dehydrogenase; Gad, gluconate dehydrogenase; Glk, glucokinase; GnuK, gluconokinase; KguK, 2-KG kinase; Zwf, glucose-6-phosphate 1-dehydrogenase; Pgl, 6-phosphogluconolactonase; KguD, 2-KG reductase; Edd, phosphogluconate dehydratase; Eda, 2-keto-3-deoxy gluconate aldolase; Glpd, glycerol-3-phosphate dehydrogenase; GapA, glyceraldehyde-3-phosphate dehydrogenase; Pyk, pyruvate kinase; AceEF, pyruvate dehydrogenase. KDPG, 2-keto-3-deoxy-6-phosphogluconate; G3P, glyceraldehyde-3P; 3PG, 3-phosphoglycerate; PEP, phosphoenolpyruvate. (**B**) The growth curve of *P. aeruginosa* strains cultured in 96-well plates containing minimal medium (MM) supplemented with 10 mM glucose as the sole carbon source. PAO1, Δ*gtrS-gltR*, Δ*gcd*, and Δ*gcd*Δ*gtrS-gltR* harbor an empty pAK1900 vector as control; p-*gtrS-gltR* denotes the pAK1900-*gtrS-gltR* plasmid; p-*gcd* denotes the pAK1900-*gcd* plasmid. OD_600_, an optical density at 600 nm. Data from n = 3 biological replicates reported as mean ± SD.

**Figure 2 microorganisms-09-00447-f002:**
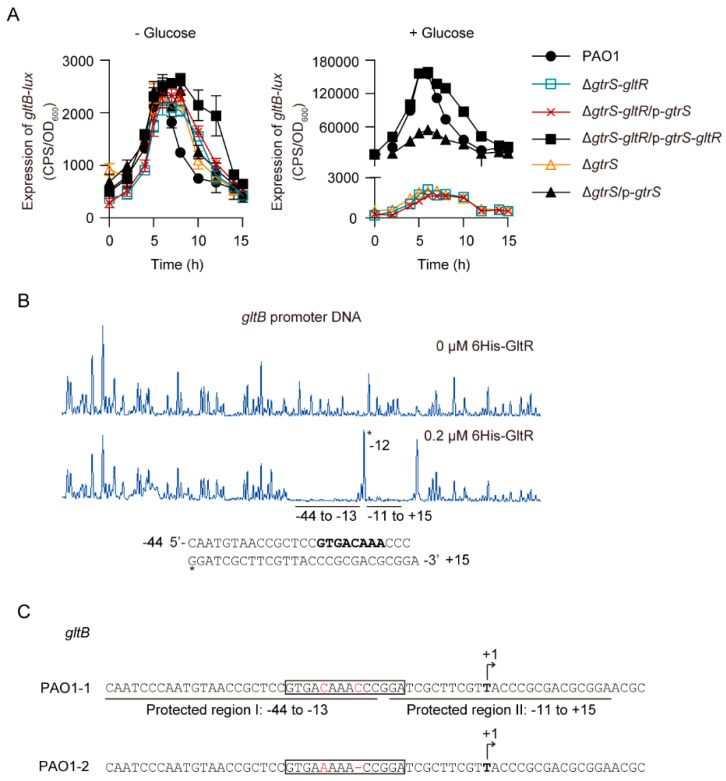
GtrS-GltR TCS positively controls the promoter activity of *gltB*. (**A**) The expression of *gltB-lux* in *P. aeruginosa* strains cultured in 96-well plates containing M8-MM supplemented with (+) or without (−) 10 mM glucose. CPS, counts per second. OD_600_, an optical density at 600 nm. Data from n = 3 biological replicates reported as mean ± SD. PAO1, Δ*gtrS-gltR*, and Δ*gtrS* harbor an empty pAK1900 vector as control; p-*gtrS-gltR* denotes the pAK1900-*gtrS-gltR* plasmid; p-*gtrS* denotes the pAK1900-*gtrS* plasmid. (**B**) Electropherograms show the protection pattern of the *gltB* promoter DNA after digestion with DNase I following incubation in the absence or presence of 6His-GltR. The protected region (relative to the transcriptional initiation site of *gltB*) was underlined, and the asterisk indicates the DNase I hypersensitivity site. The sequence covering the two GltR-protected regions in the *gltB* promoter DNA (*gltB-p*) are shown. The conserved GltR-binding site inferred from previous study by Daddaoua et al. (2014) [26] is in bold. (**C**) Analysis of the DNA sequence of *gltB* promoter in *P. aeruginosa* strains. PAO1-1, partial *gltB* promoter sequence of the PAO1 strain with published reference sequence [13] or that of our laboratory PAO1 strain [19]; PAO1-2, the partial *gltB* promoter sequence of the PAO1 strain used by Daddaoua et al. (2014) [26]. Boxes highlight the reported GltR-binding sites Daddaoua et al. (2014) [26] and the differences between PAO1-1 and PAO1-2 are shown with red letters. The transcriptional start sites inferred from Daddaoua et al. (2014) [26] are indicated by arrows and bold letters.

**Figure 3 microorganisms-09-00447-f003:**
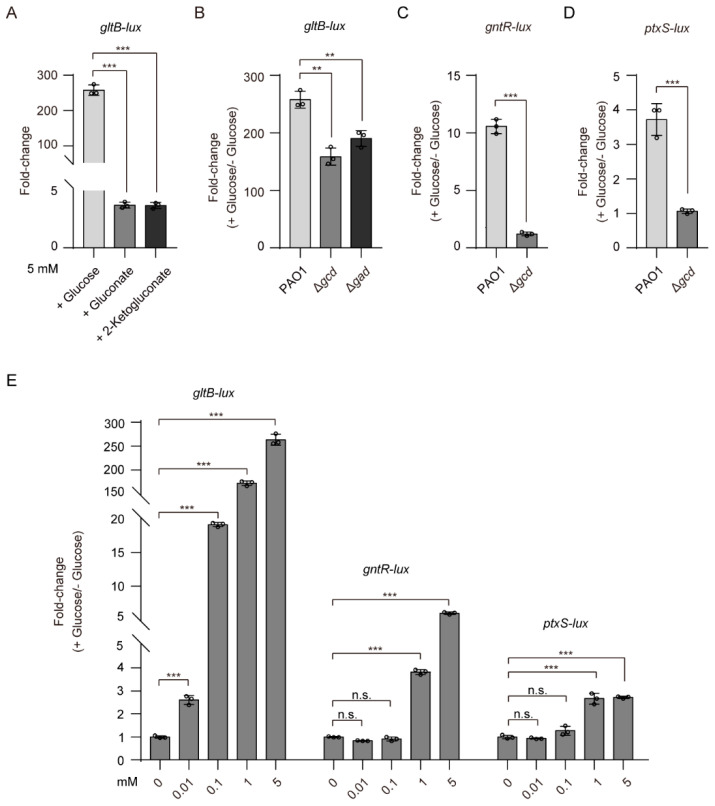
Expression of glucose utilization and metabolism genes. (**A**) Effect of glucose, gluconate, and 2-ketogluconate on the expression of *gltB-lux* in WT PAO1 strain grown in tubes containing M9 MM at 37 °C with shaking (250 rpm) for 8 h. (**B**) Effect of glucose on the expression of *gltB-lux* in *P. aeruginosa* PAO1 and its derivatives grown in tubes containing M9 MM at 37 °C with shaking (250 rpm) for 8 h. (**C**,**D**) Effect of glucose on the expression of *gntR-lux* (C) and *ptxS-lux* (D) in *P. aeruginosa* strains grown in tubes containing M9 MM at 37 °C with shaking (250 rpm) for 8 h. (**E**) Effect of different concentrations of glucose on the expression of *gltB-lux*, *gntR-lux*, and *ptxS-lux* in PAO1 cultured in tubes containing M9 MM at 37 °C with shaking (250 rpm) for 8 h. In (A to E), data from n = 3 biological replicates reported as mean ± SD. (** *p* < 0.01, *** *p* < 0.001, n.s. indicates no significant difference; Student’s two-tailed *t*-test).

**Figure 4 microorganisms-09-00447-f004:**
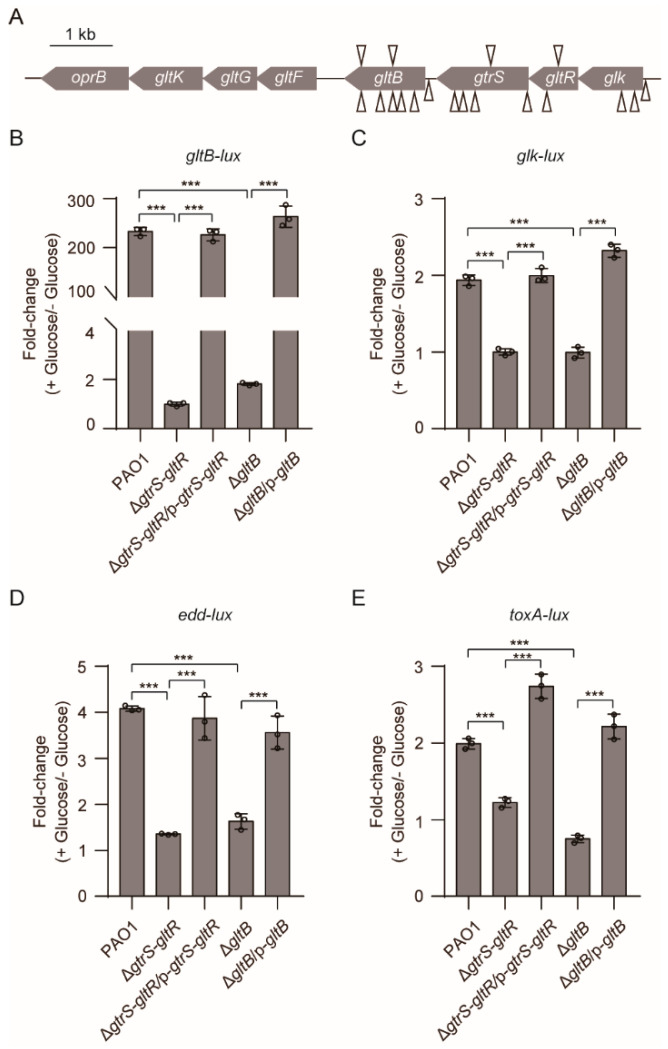
Disruption of *gltB* decreases the expression of GltR-regulated genes in response to glucose. (**A**) Schematic representation of the transposon insertion site. Gene is shown by arrow and the triangle indicated the mariner-based transposon. The direction (relative to the genomic scaffold) of gentamicin resistance cassette in transposon is provided (forward, upper triangles; reverse, lower triangles). (**B**–**E**) The effect of either *gtrS-gltR* or *gltB* deletion on the glucose-induced expression of known GltR–regulated genes. Bacteria were grown in tubes containing M9 MM supplemented with (+) or without (−) 5 mM glucose at 37 °C with shaking (250 rpm) for 8 h, and the fold-change in gene expression (induced by glucose) was determined. Data from n = 3 biological replicates reported as mean ± SD. (*** *p* < 0.001; Student’s two-tailed *t*-test). PAO1, Δ*gtrS-gltR, and* Δ*gltB* harbor an empty pAK1900 vector as control; p-*gtrS-gltR* denotes the pAK1900-*gtrS-gltR* plasmid; p-*gltB* denotes the pAK1900-*gltB* plasmid.

**Figure 5 microorganisms-09-00447-f005:**
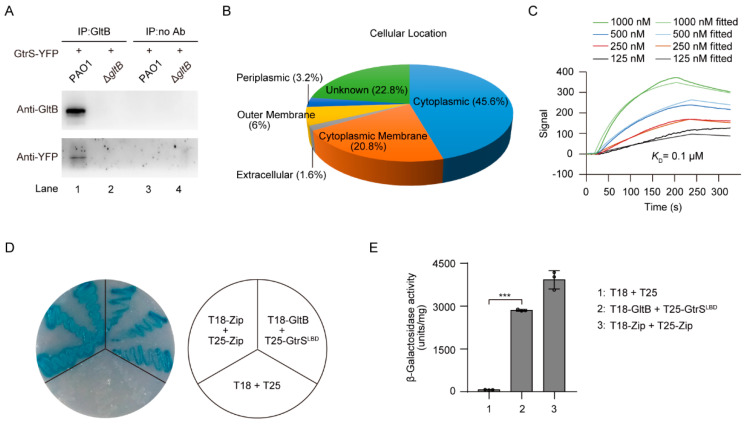
GltB can form a complex with GtrS. (**A**) Western blot image showing the co-Immunoprecipitation of GltB and GtrS-YFP. Whole cell extracts from WT PAO1 and Δ*gltB* mutant strains expressing *gtrS*-YFP were immunoprecipitated with (lanes 1, 2) or without (lanes 3, 4) anti-GltB antibody. (**B**) Categories of the putative GltB-binding proteins identified by co-Immunoprecipitation coupled with mass spectrometric (CoIP-MS). The identified proteins were classified into extracellular, outer membrane, periplasmic, cytoplasmic membrane, cytoplasmic, and unknown according to their potential cellular location (http://www.pseudomonas.com/ accessed on 1 May 2020). Percentage of proteins was indicated in the graph. (**C**) Surface plasmon resonance (SPR) analysis showing the interaction of 6His-GltB with 6His-GtrS^LBD^. Analysis performed in TraceDrawer using a 1:1 binding interaction model. *K*_D_, equilibrium dissociation constant. (D and E) Bacterial two-hybrid (BACTH) assays assessing the interaction of GltB and GtrS^LBD^. In (**D**), *E. coli* BTH101 recombinants bearing indicated combinations of plasmids were plated on selective media with maltose as the unique carbon source, and incubated at 30 °C for 6 days; in (**E**), the expression of β-galactosidase activity was examined in *E. coli* BTH101 recombinants bearing indicated combinations of plasmids cultured in LB broth containing 0.5 mM isopropyl-1-thio-β-d-galactopyranoside (IPTG) for 12 h, and data represented mean ± SD from n = 3 biological replicates (*** *p* < 0.001; Student’s two-tailed *t*-test). T18C/T25 (negative control plasmid), T18C-zip/T25-zip (positive control plasmid).

**Figure 6 microorganisms-09-00447-f006:**
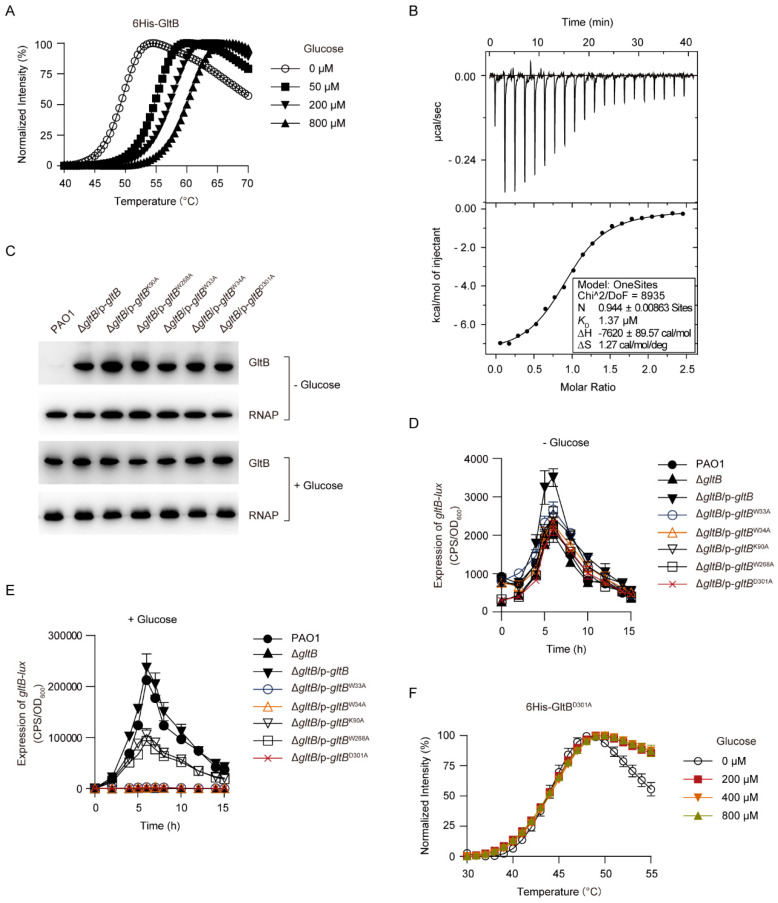
Role of glucose-binding residues on the regulatory function of GltB. (**A**) Thermal melt curves of 5 μM 6His-GltB in the presence of indicated glucose concentrations. (**B**) Isothermal titration calorimetry of 6His-GltB (17 μM) with glucose (200 μM). Upper panel, raw data output; lower panel, plot of integrated heats versus the glucose/6His-GltB ratio. Data was fitted with a one-site model. The fitted results are displayed: N refers to the stoichiometry of GltB-glucose complex, *K*_D_ refers to the equilibrium dissociation constant, Δ*H* (reaction enthalpy) and Δ*S* (entropy) are the thermodynamic parameters. (**C**) Western blot images showing the production of GltB and its variants. Protein samples were derived from bacteria grown in tubes containing M8 MM in the absence (−) or presence (+) of 10 mM glucose at 37 °C for 6 h with shaking (250 rpm). RNAP subunit is probed as a loading control. Experiments were repeated at least three times with similar results and the figures show a set of representative data. (**D**,**E**) Expression of *gltB-lux* in *P. aeruginosa* strains cultured in 96-well plates containing M8 MM supplemented with (+) or without (−) 10 mM glucose. Data from n = 3 biological replicates reported as mean ± SD. In (D and E), PAO1 and Δ*gltB* harbor an empty pAK1900 vector as control; p-*gltB* denotes the pAK1900-*gltB* plasmid; p-*gltB*^W33A^, p-*gltB*^W34A^, p-*gltB*^K90A^, p-*gltB*^W268A^, and p-*gltB*^D301A^ denote the pAK1900-*gltB*^W33A^ plasmid, the pAK1900-*gltB*^W34A^ plasmid, the pAK1900-*gltB*^K90A^ plasmid, the pAK1900-*gltB*^W268A^ plasmid, and the pAK1900-*gltB*^D301A^ plasmid, respectively (Appendix A). (**F**) Thermal melt curves of 5 μM 6His-GltB^D301A^ in the presence of indicated glucose concentrations. In (A) and (F), fluorescence signals of all samples have been normalized to relative values of 0% (the lowest fluorescence signal) and 100% (the highest fluorescence signal), respectively.

**Figure 7 microorganisms-09-00447-f007:**
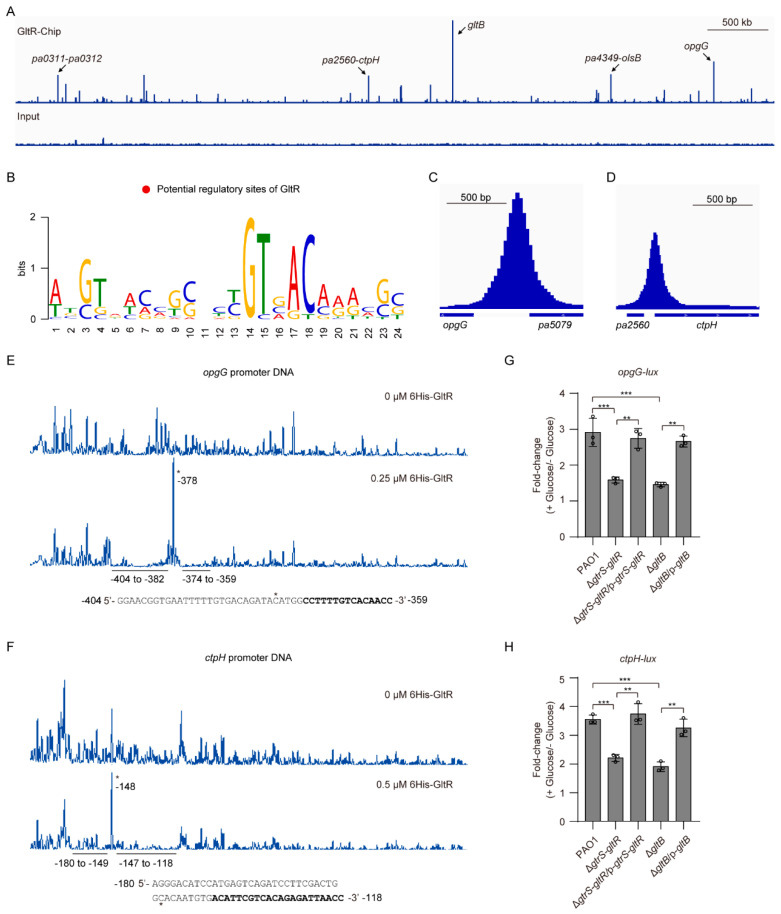
Genome-wide identification of GltR targets. (**A**) Images showing GltR-binding sites identified by CHIP-seq. The top five of CHIP-seq peaks were labeled. (**B**) The most significant motif (E-value = 1.3 × 10^−6^) identified by MEME tool was observed in all DNA sequences of the 16 tested peaks (≥10-fold enrichment, in average). The height of each letter represents the relative frequency of each base at different position in the consensus sequence. (**C**,**D**) Pattern of CHIP-seq peaks identified in intergenic regions of *opgG-pa5079* (C) and *pa2560*-*ctpH* (D). (E and F) Electropherograms show the protection patterns of the *opgG* (**E**) and *ctpH* (**F**) promoter DNA after digestion with DNase I following incubation in the absence or presence of 6His-GltR. The protected region (relative to the start codon) is underlined, and the asterisk indicates the DNase I hypersensitivity site. The sequences including GltR-protected regions are shown, and the conserved sequence identified by MEME of GltR-binding sites are shown in bold. (G and H) Effect of either *gtrS-gltR* or *gltB* deletion on the expression of *opgG-lux* (**G**) and *ctpH-lux* (**H**). Bacteria were grown in tubes containing M9 MM supplemented with (+) or without (−) 5 mM glucose at 37 °C with shaking (250 rpm) for 8 h. Data from n = 3 biological replicates reported as mean ± SD. (** *p* < 0.01, *** *p* < 0.001; Student’s two-tailed *t*-test). PAO1, Δ*gtrS-gltR,* and Δ*gltB* harbor an empty pAK1900 vector as control; p-*gtrS-gltR* denotes the pAK1900-*gtrS-gltR* plasmid; p-*gltB* denotes the pAK1900-*gltB* plasmid.

**Figure 8 microorganisms-09-00447-f008:**
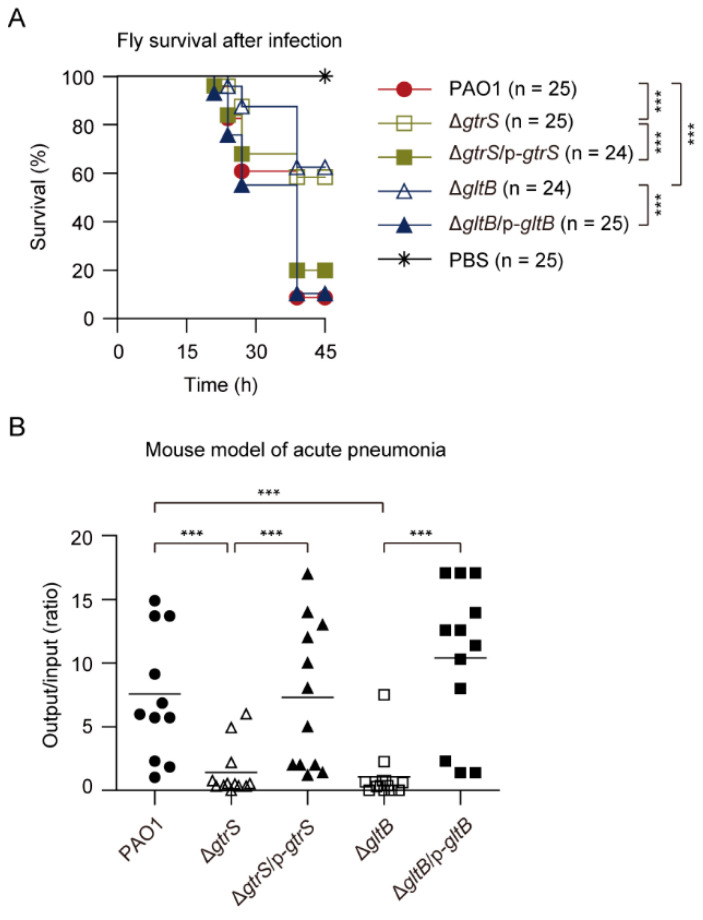
Role of GltB in *P. aeruginosa* virulence. (**A**) Survival rates of fly upon infection by indicated *P. aeruginosa* strains. n indicates the number of flies used. Asterisks denote statistical significance determined by log-rank Test: *** *p* < 0.001. PBS indicates flies was pricked with a sterilized tungsten needle dipped in phosphate buffer saline (PBS) buffer. (**B**) Recovery of *P. aeruginosa* in a mouse model of acute pneumonia. Results are expressed as the ratio of CFU (colony-forming unit) recovered per lung (output) to CFU present in the initial inoculum (input). Data represent results from n = 10–12 mice per strain; the line shows the geometric mean for each group. The Mann–Whitney test was used to calculate p-values (two-tailed): *** *p* < 0.001. In all panels, PAO1, Δ*gtrS*, and Δ*gltB* harbor an empty pAK1900 vector as control; p-*gtrS* denotes the pAK1900-*gtrS* plasmid; p-*gltB* denotes the pAK1900-*gltB* plasmid.

## Data Availability

Data available in a publicly accessible repository.

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
