# Peer review of "Glucose-Binding of Periplasmic Protein GltB Activates GtrS-GltR Two-Component System in Pseudomonas aeruginosa"

_microorganisms, 2021, doi:10.3390/microorganisms9020447_

Round 1
Reviewer 1 Report
In this superb manuscript, Xu and coworkers define the regulatory mechanisms for the two-component system (TCS) GtrS-GltR. The authors first show that the GtrS-GltR system is required for growth on glucose when Gcd is absent, demonstrating overlapping function between these glucose sensing mechanisms. The authors use robust reporter assays to show that this TCS positively regulates the activity of the gltBFGK-oprB operons. EMSA gel-shift assays and DNAse I footprinting were used to define the site of response regulator binding. The authors then undertake a mutant hunt in a comprehensive transposon mutant library to identify GltB, a known glucose-binding periplasmic protein, as a factor required for transmitting the glucose signal to the TCS. The glucose-binding pocket is demonstrated to be required for signal transmission. Genome-wide ChIP Seq experiments identify myny new targets subject to GltR regulation. Finally the TCS is shown to be required for full virulence in mouse and fly models of infection.
I have little to recommend for improvement of this manuscript. The methods used in this study are all standard approaches and are robust in nature. The writing could be improved with more careful attention to proper tense. The authors are to be commended for their careful consideration on the interpretation of the GltB binding parters (there is often a long list of interactors in these types of experiments when dealing with periplasm proteins).
Reviewer 2 Report
In this manuscript, Xu et al. describe the function of the periplasmic protein GltB and the GtrS-GltR two component system in glucose utilization in Pseudomonas aeruginosa (Pa). The authors have present fantastic data showing the interplay of these three proteins and their regulatory effect on Pa metabolism and virulence. The data presented herein is novel and further expands our knowledge of the importance of glucose metabolism to Pa virulence. The conclusion presented by the authors are supported by the data. This manuscript will have high impact and deserves publication. Only a very few minor comments are listed below:
Fig S6: This figure in my opinion is important. If space and guidelines permit, then it should be part of the main text instead as a supplementary figure.
Line 75: Change ‘showing’ to show
Lines 118-119: Is this phenotype also the same in the absence of gltR, the response regulator of the TCS?
Fig. S3A. Why is it that complementing the gtrS mutant with the gtrS-yfp fusion (inverted triangle) restores wt levels of promoter activity, but complementing with just gtrS (regular triangle) only partially restores promoter activity?
Lines 360-375, Fig 6 data suggests that GltB and the TCS are two redundant pathways required for activation of opgG and ctpH. It would have been great to see if a gltB gtrS gltR triple mutant reduces expression of either opgG and ctpH even more than the single and double mutants.
Reviewer 3 Report
The paper is interesting, the topic is highly specific and correctly contextualized. The text is in general well written, but it requires some revisions.
In the section of results there are several references. This greatly distracts the reader from the logical rationale of the results. The results section should be focused only to the report of results, unless you decide to make a section integrating results and discussions. Please, adjust this section according to the journal guidelines.
Another point is that a lot of references are really old. I suggest to update the reference to make more appreciable the paper.
Some details below:
Line 75: showing or showed?
Lines 652-657. This sentence is too long, please split it.
Lnes 654-655. Supplemented without is not a correct expression.
Lines 673-677. Too long sentence, please split it.
Lines 681-689. I think this paragraph could be synthesized. If the general procedure is the same for all the carbon sources you can condense the whole paragraph.
